# Glycan-mediated enhancement of reovirus receptor binding

Melanie Koehler [1,5], Pavithra Aravamudhan[2,3,5], Camila Guzman-Cardozo[2,3], Andra C. Dumitru[1], Jinsung Yang[1], Serena Gargiulo[1], Patrice Soumillion[1], Terence S. Dermody[2,3,4]* & David Alsteens [1]*

Viral infection is an intricate process that requires the concerted action of both viral and host cell components. Entry of viruses into cells is initiated by interactions between viral proteins and their cell surface receptors. Despite recent progress, the molecular mechanisms underlying the multistep reovirus entry process are poorly understood. Using atomic force microscopy, we investigated how the reovirus σ1 attachment protein binds to both α-linked sialic acid (α-SA) and JAM-A cell-surface receptors. We discovered that initial σ1 binding to α-SA favors a strong multivalent anchorage to JAM-A. The enhanced JAM-A binding by virions following α-SA engagement is comparable to JAM-A binding by infectious subvirion particles (ISVPs) in the absence of α-SA. Since ISVPs have an extended σ1 conformer, this finding suggests that α-SA binding triggers a conformational change in σ1. These results provide new insights into the function of viral attachment proteins in the initiation of infection and open new avenues for the use of reoviruses as oncolytic agents.

[1] Louvain Institute of Biomolecular Science and Technology, Université catholique de Louvain, Louvain-la-Neuve, Belgium. [2] Department of Pediatrics, University of Pittsburgh School of Medicine, Pittsburgh, PA, USA. [3] Center for Microbial Pathogenesis, UPMC Children's Hospital of Pittsburgh, Pittsburgh, PA, USA. [4] Department of Microbiology and Molecular Genetics, University of Pittsburgh School of Medicine, Pittsburgh, PA, USA. [5] These authors contributed equally: Melanie Koehler, Pavithra Aravamudhan. *email: terence.dermody@chp.edu; david.alsteens@uclouvain.be

V iruses are strict intracellular pathogens that depend on host organisms for virtually all stages of their replication cycle[1]. During millions of years of evolution and adaptation to their hosts, viruses acquired the relevant molecular factors to exploit and control cellular functions[2]. Receptor-mediated virus entry into host cells is a complex multistep process in which viruses must overcome a variety of obstacles to access host machinery for replication[3]. Virus-cell-surface interactions determine mechanisms of virus attachment, uptake, intracellular trafficking, and ultimately, penetration into the cytosol. Elucidating the complex interplay of viruses and their receptors is important to gain a full understanding of the entry process. While tremendous efforts have been made to define the cellular receptors and the entry pathways that mediate virus internalization, our current knowledge of virus entry relies mainly on ensemble studies that provide an average measurement of a population of viruses or on static imaging observations using fixed cells. However, virus uptake by cells is often stochastic, suggesting that the average does not account for biological variability[4]. Because virus infection is a multistep process in which the dynamics of each individual step are crucial, conducting experiments using living cells maintained under physiological conditions is essential[3].

Mammalian orthoreoviruses (reoviruses) are nonenveloped, double-stranded RNA viruses that assemble virions consisting of two concentric protein shells, outer capsid and core. While not recognized as common human pathogens, a recent study supports a role for reovirus infection in triggering the development of celiac disease by breaking immunological tolerance to orally ingested gluten[5]. In addition, reovirus efficiently lyses tumor cells[6,7] and has shown efficacy in clinical trials against refractory human cancers[8–10]. The reovirus outer-capsid protein σ1 engages host cell-surface factors to enable cell entry (Fig. 1a)[11]. Protein σ1 is a fibrous trimer that consists of two domains, an elongated tail domain that inserts at its base into the viral particle and a globular head that projects distally from the virion surface (Fig. 1b). Both σ1 tail and head domains in serotype 3 reovirus contain receptor-binding sequences. The σ1 tail domain binds α-linked sialic acid (α-SA), whereas the head domain binds junctional adhesion molecule A (JAM-A) (Fig. 1b, c)[12]. A single point

mutation in the σ1 tail region implicated in α-SA binding influences the neurovirulence of serotype 3 reovirus[13]. JAM-A serves as a receptor for all three reovirus serotypes[11,14,15]. However, the dynamics of reovirus binding to these two cell-surface receptors have not been elucidated in the context of living cells.

Here, we analyze the binding of reovirus in vitro and on living cells in order to dissect the respective contributions of the α-SA and JAM-A receptors to stable cell attachment. We extract the kinetics and thermodynamics of virus-receptor interactions in vitro and analyze the capacity of reovirus to form multivalent interactions. This in vitro calibration allowed us to quantify the number of bonds establish at the single-virion level directly on living cells. Surprisingly, we discovered that initial σ1 binding to α-SA acts as a trigger that enhances the overall avidity of σ1 for JAM-A, binding to which is a critical step in viral entry. These findings provide evidence for the interplay between a viral attachment protein and specific host receptors during viral attachment to the cell surface and open new avenues to better control reovirus infection in the context of live attenuated reovirus vaccines or the use of reovirus as an oncolytic agent.

## Results

**σ1 attaches to α-SA glycans through multivalent bonds.** As σ1 binding to α-SA glycans is the first step in reovirus attachment to the cell surface[15], we used atomic force microscopy (AFM) (the principle of force-distance-based AFM is described in Supplementary Fig. 1) to evaluate the binding strength of reovirus to α-SA using both model surfaces (Supplementary Fig. 2 validates the reovirus virion morphology, tip functionalization, and model surface chemistries) and living cells (Supplementary Fig. 3 describes the cell lines used). To mimic cell-surface glycans in vitro, biotinylated-α-SA glycans were immobilized onto streptavidin-coated surfaces to allow virus access to α-SA[16,17]. These model surfaces were imaged by AFM, and the thickness of the grafted layer was validated by a scratching experiment, revealing a deposited layer of ~1.0 ± 0.3 nm (see methods and Supplementary Fig. 2d, cross-section in inset). To study reovirus binding to α-SA, we covalently attached purified virions of T3SA+ (Supplementary Fig. 2c, showing single virions at the tip

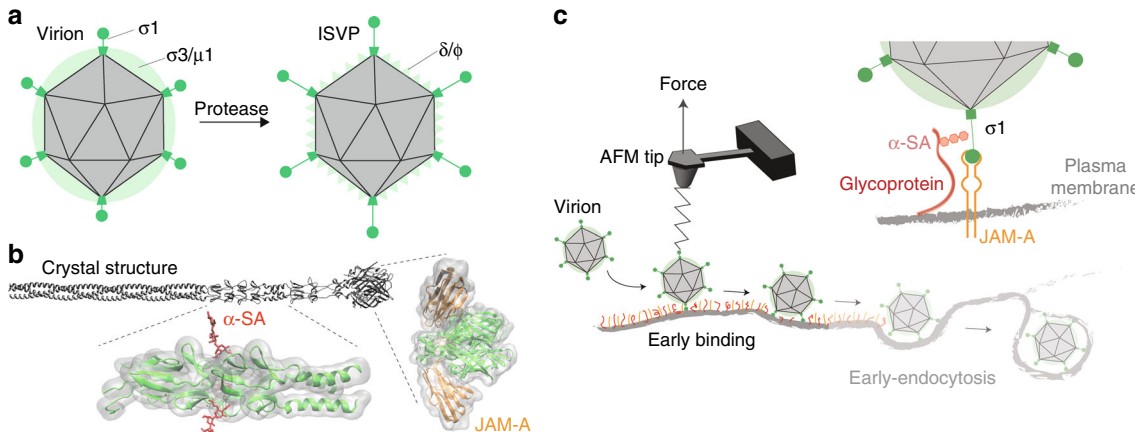

**Fig. 1** Probing reovirus binding to living cells. **a** Schematic of reovirus particles with outer-capsid proteins labeled before (virion) and after (infectious subvirion particle [ISVP]) proteolytic processing. The cartoon shows the arrangement of outer-capsid proteins in the double-layered shell of virions, the formation of ISVPs by removal of σ3 and cleavage of μ1 to yield δ and φ, and rearrangement of σ1 into a more elongated conformation. **b** Full-length model of reovirus σ1 protein[28], which functions as the viral attachment protein that binds to cell-surface glycans (in particular, to terminal α-linked sialic acid [α-SA] residues) and junctional adhesion molecule-A (JAM-A). Regions of the molecule that interact with α-SA and JAM-A are indicated. **c** Schematic of probing reovirus entry using AFM. The initial attachment of reovirus to cells involves specific binding between the viral σ1 protein and the receptor, JAM-A. Cell-surface glycans serve as attachment factors, and virus binding to their α-SA groups function in the initial association of virus to cells and further facilitates high-affinity binding to JAM-A

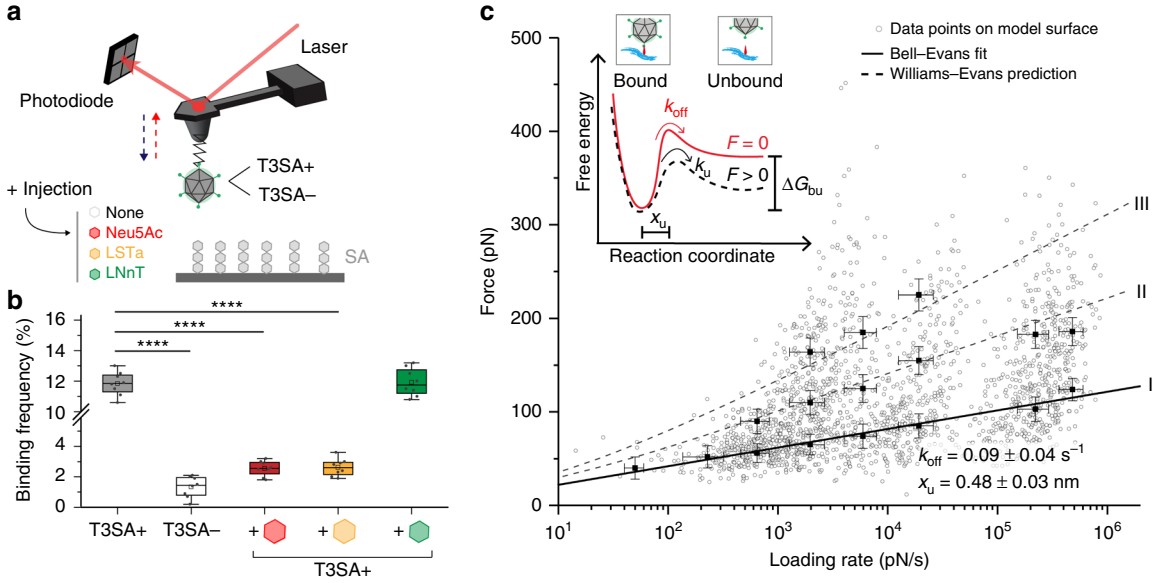

**Fig. 2** Probing T3 reovirus binding to sialylated glycans on model surfaces. **a** Binding of single virions is probed on an SA-coated surface in the presence or absence of α-SA glycan derivatives: N-acetylneuraminic acid (Neu5Ac), sialyl-lacto-N-tetraose a (LSTa), and a derivative without α-SA (lacto-N-neotetraose [LNnT]). **b** Box plot of specific binding frequencies measured by AFM between virions and α-SA before and after injection of 1 mM glycans. The horizontal line within the box indicates the median, boundaries of the box indicate the 25th and 75th percentile, and the whiskers indicate the highest and lowest values of the results. The square in the box indicates the mean. **c** Dynamic force spectroscopy (DFS) plot showing the distribution of rupture forces measured between T3SA+ and the SA-coated surface (gray dots) with average rupture forces determined for eight distinct loading rate (LR) ranges. Data corresponding to single interactions are fitted with the Bell-Evans (BE) model describing a ligand-receptor bond as a simple two-state model (I, black curve). Dashed lines represent the predicted binding forces for two (II) and three (III) simultaneous uncorrelated interactions (Williams-Evans model [WEM]). Inset: BE model describing a ligand-receptor bond as a simple two-state model. The bound state is separated from the unbound state by an energy barrier located at distance $x_u$. $k_u$ and $k_{off}$ represent the transition rate and transition at thermal equilibrium, respectively. Error bars indicate s.d. of the mean value. For all experiments, data are representative of $n = 5$ independent experiments. ****$P < 0.0001$; determined by two-sample $t$-test in Origin. Source data are provided as a Source Data file

apex), a wild-type reovirus strain, to the free end of a long, polyethylene glycol (PEG)$_{27}$ spacer chemically linked to the AFM tip[18,19]. Force-distance (FD) curves were recorded to assess the binding strength between T3SA+ virions and α-SA glycans (Fig. 2). Specific adhesion events were observed on 10–15% of retraction FD curves at rupture distances >5 nm, which corresponds to the extension of the PEG linker. To confirm the specificity of these interactions, we conducted additional independent control experiments using (i) an AFM tip functionalized with a mutant reovirus strain, T3SA−, which does not engage α-SA by virtue of a P204L mutation in the α-SA-binding site in the tail domain of σ1[20] and (ii) competition assays with soluble α-SA molecules including acetylneuraminic acid (Neu5Ac), sialyl-lacto-N-tetraose (LSTa), or lacto-N-neotetraose (LNnT), a glycan lacking α-SA. As expected, T3SA− did not display significant binding to α-SA, and the injection of free Neu5Ac and LSTa but not LNnT strongly competed with T3SA+ binding to α-SA (Fig. 2b). These controls confirm the specificity of interactions and the critical importance of specific residues in the σ1 tail domain for α-SA binding.

To extract the kinetic parameters describing the σ1-α-SA interaction, we force-probed the interactions at various force loading rates[21] (Fig. 2c, Supplementary Fig. 1c, d). Using the physiologically relevant direction-of-force application, the σ1-α-SA complex withstood forces in the range of 25–400 pN. This force regime is usually associated with the stability of the protein conformation, raising the concern that the reovirus virions linked to the AFM tip could be damaged over time. Because the apices of the cantilevers have radii of ~40 nm, they only can host a few viral particles, as evidenced by laser-scanning optical microscopy (Supplementary Fig. 2c). If the reovirus virions at the tip apex

were mechanically altered, such alterations would produce a rapid decrease in the frequency of interactions over time. In contrast, a single cantilever remained active over thousands of interactions and several scans, indicating that tip and surface functionalization can sustain such high forces.

According to the Bell-Evans (BE) model[22,23], the σ1-α-SA interaction can be described as a simple two-state model, in which the bound state is separated from the unbound state by a single energy barrier located at distance $x_u = 0.48 \pm 0.03$ nm and crossed with a transition rate of $k_{off} = 0.09 \pm 0.04$ s$^{-1}$. We also observed bivalent and trivalent interactions. These multivalent interactions appear as uncorrelated bonds loaded in parallel, as confirmed by the predictive Williams-Evans (WE) model[24] (Fig. 2c, dashed curves II and III). These multivalent interactions are most likely established between σ1 molecules on a single virion attached to the AFM tip and multiple α-SA molecules immobilized on the surface. This hypothesis is supported by the following reasons: (i) σ1 is a trimer with three binding sites; (ii) each virion possess multiple copies (up to 12, corresponding to the virion icosahedral vertices) of the σ1 trimer; (iii) the tip apex bears only one or two virions; and (iv) the unbinding occurs in a single step (a single rupture peak observed in the FD curves). Thus, our in vitro experiments confirm that T3SA+ virions specifically interact with α-SA glycans and that virions rapidly (in the ms range) establish multivalent bonds with α-SA glycans, presumably providing the virion with its first stable anchorage to the cell surface.

We next conducted assays using living CHO cells that express α-SA (cells fluorescently labeled with a nuclear protein H2B-eGFP and actin-mCherry) and Lec2 cells deficient in α-SA (~70–90% deficiency of SA in their glycoproteins and

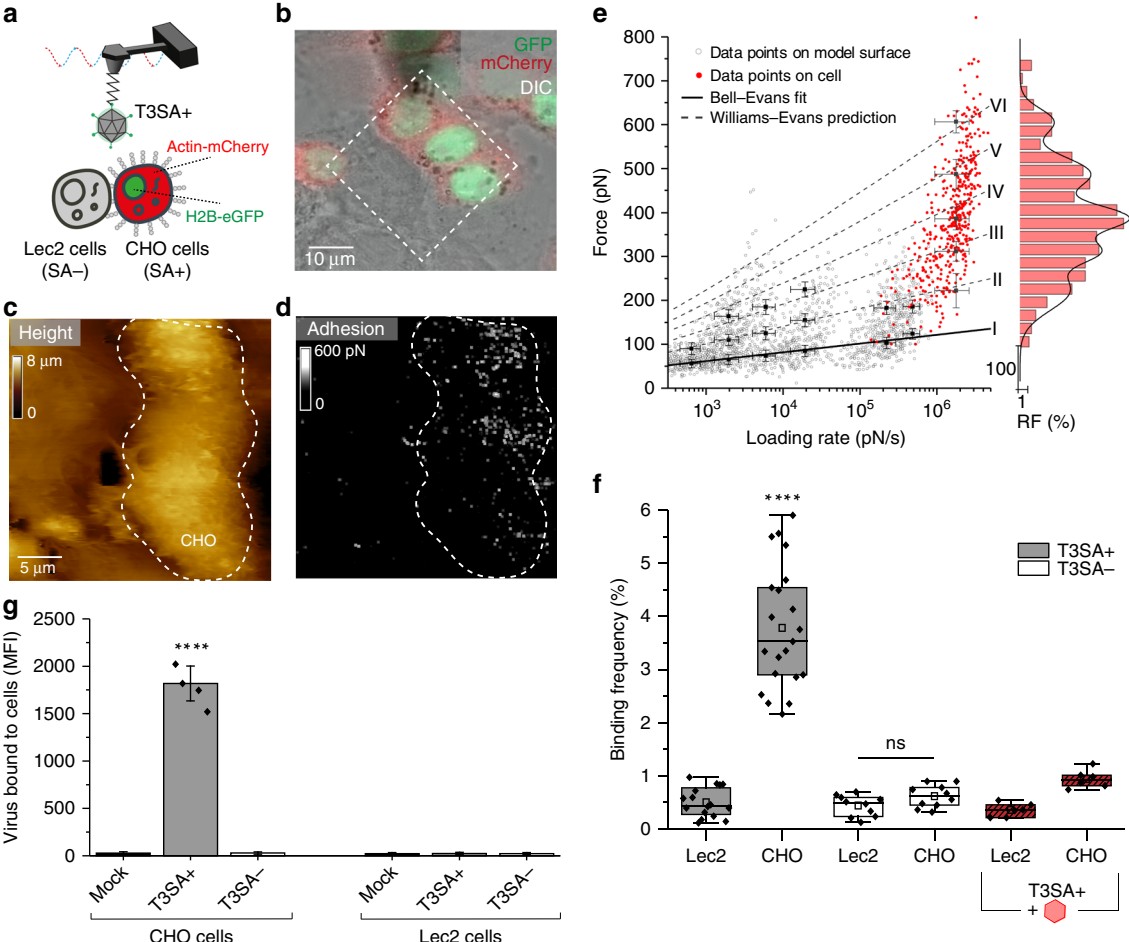

**Fig. 3** Probing T3 reovirus binding to sialylated glycans on living cells. **a** Combined optical microscopy and FD-based AFM of T3SA+ binding to cells expressing (CHO) or lacking (Lec2) α-SA on the cell surface. **b** Overlay of DIC, eGFP, and mCherry signals of a confluent layer of co-cultured fluorescent CHO cells (actin-mCherry and H2B-eGFP) and Lec2 cells. **c, d** FD-based AFM topography image (**c**) and corresponding adhesion map (**d**) from probing of adjacent cells indicated in the dashed square in **b**. The adhesion map shows interactions mainly on CHO cells (α-SA-expressing cells) (white pixels). For higher visibility, the pixel size in the adhesion image was enlarged two-fold. **e** DFS plot of data from α-SA model surfaces (grey circles, from Fig. 2c) and living cells (red dots). Histogram of the force distribution observed on cells fitted with a multi-peak Gaussian distribution (n = 700 data points) is shown at the side. **f** Box plot of BF observed for T3SA+ (gray) and T3SA− (white) virions as well as T3SA+ virions following injection of 1 mM Neu5Ac (red). The horizontal line within the box indicates the median, boundaries of the box indicate the 25th and 75th percentile, and the whiskers indicate the highest and lowest values of the results. The square in the box indicates the mean. For all experiments, data are representative of at least n = 5 independent experiments. **g** Influence of SA on virus binding determined by flow cytometry. Cells were incubated with either PBS (Mock) or Alexa Flour 488-labeled T3SA+ or T3SA− virions (10⁵ particles per cell), and the median fluorescence intensity (MFI) of cell-bound virus was determined by flow cytometry as shown in Supplementary Fig. 3a. Error bars indicate s.d. of the mean value. Experiments were repeated twice (n = 2 independent experiments, each with duplicate samples). ns, P > 0.05; ****P < 0.0001; determined by two-sample t-test in Origin (**f**) and by two-way ANOVA corrected for multiple comparisons using Tukey's test in GraphPad Prism (**g**), respectively. Source data are provided as a Source Data file

gangliosides [Supplementary Fig. 3b, c])[25]. Using AFM tips functionalized with T3SA+, we imaged a confluent monolayer of co-cultured CHO and Lec2 cells using conditions to propagate both cell types[18] (Fig. 3a–d). Guided by fluorescence intensity, we chose fields of view in which both cell types were adjacent, serving as a direct internal control in AFM imaging (Fig. 3b). AFM height images were recorded together with corresponding adhesion images, revealing the location of specific adhesion events displayed as bright pixels on the adhesion map (Fig. 3c, d). Remarkably, CHO cells showed a high density of adhesion events (~4%, Fig. 3d, f), whereas Lec2 cells displayed only a sparse distribution of these events (<1%, Fig. 3d, f), suggesting the establishment of specific T3SA+-α-SA bonds on living cells. We also assessed the stability of the virions on the AFM tip apex by recording consecutive maps with the same T3SA+ tip. The presence of virus on the AFM tip during consecutive maps

excludes the possibility of virions becoming internalized during our AFM experiments (Supplementary Fig. 4a–d). In fact, due to the low contact time (in the ~ms range), the probability of observing internalization events is extremely low. Accordingly, reovirus virions are essentially stationary during the early stage of binding to the cell surface (between 280 and 1500 s)[26]. The rupture forces (Fig. 3e, red dots) were in the range of 50-400 pN. Using the WE prediction (established from in vitro data), we deduce that T3SA+ virions establish up to six interactions in parallel with a maximum likelihood of three-to-four interactions (Fig. 3e, dashed lines [II to VI] and histogram). These results suggest that T3SA+ virions, despite a brief contact time of ~1 ms with the cell surface, are capable of forming multiple interactions in parallel. As the σ1 protein forms trimers that theoretically can interact simultaneously with up to three α-SA glycans, these results suggest that the virion uses more than a single σ1 protein

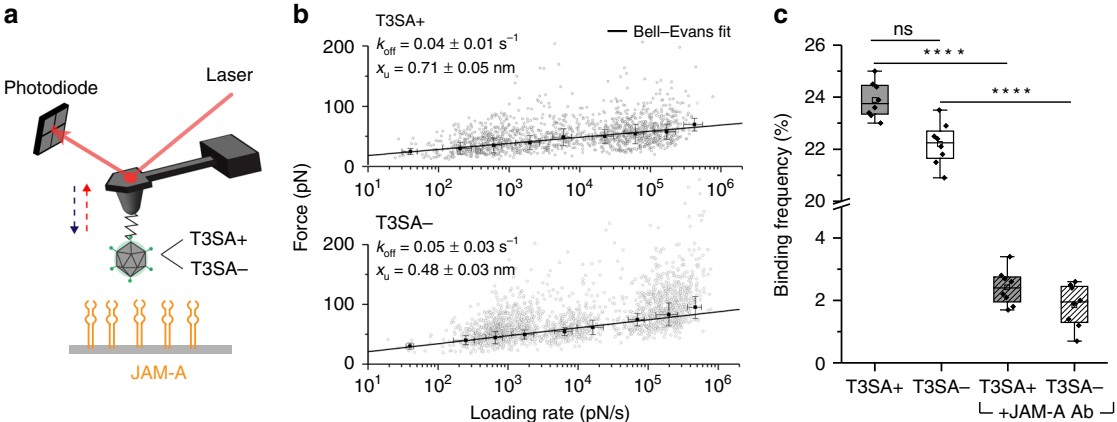

**Fig. 4** Probing reovirus binding to JAM-A on model surfaces. **a** Binding of single virions (T3SA+ or T3SA−) to JAM-A probed on a model surface. **b** DFS plot showing the force required to separate T3SA+ (upper panel) or T3SA− (lower panel) virions from JAM-A and fitted with the BE model. Error bars indicate s.d. of the mean value. For all experiments, data are representative of at least $n = 5$ independent experiments. **c** Box plot of the binding frequency of reovirus to JAM-A model surface. T3SA+ interactions are shown in grey, T3SA− in white, and hatched boxes represent injection of 10 µg/mL JAM-A antibody (Ab). The horizontal line within the box indicates the median, boundaries of the box indicate the 25th and 75th percentile, and the whiskers indicate the highest and lowest values of the results. The square in the box indicates the mean. ns, $P > 0.05$; ****$P < 0.0001$; determined by two-sample t-test in Origin. Source data are provided as a Source Data file

for its early attachment to the cell surface. The specificity of the T3SA+-α-SA interaction was validated using three different approaches: (i) probing the same CHO-Lec2 cell mixture first with T3SA+ on the AFM tip and then with T3SA− on the tip (Fig. 3f, Supplementary Fig. 4e–h); (ii) blocking specific virus-glycan interactions using 1 mM Neu5Ac (Fig. 3f, Supplementary Fig. 4i–l); and (iii) by flow cytometric analysis of virion binding (Fig. 3g). In the flow cytometry experiments, cells were incubated with either no virions (mock) or Alexa Fluor 488-labeled T3SA+ or T3SA− virions ($10^5$ particles per cell) for 1 h, and the median fluorescence intensity (MFI) of cell-bound virus was determined (Supplementary Fig. 3a). As shown in Fig. 3g, T3SA+ virions mainly bound to CHO cells, whereas almost no binding was detected for mock or T3SA− virions. Although binding forces appear much larger on CHO cells than on model α-SA surfaces, the specificity of the interaction demonstrated by the controls described above confirms that we are probing the same interactions on both cells and model surfaces. The larger forces are likely the result of a difference in the number of bonds established simultaneously. Together, these results confirm that T3SA+ virions establish multiple, specific interactions with α-SA glycans on living cells.

**σ1 forms stable, multivalent complexes with JAM-A receptors.** While α-SA engagement can provide the first foothold for reovirus on the cell surface, engaging a specific receptor such as JAM-A is essential for cell entry. To evaluate reovirus binding to JAM-A, we force-probed T3SA+ or T3SA− virion binding to JAM-A-coated surfaces (Fig. 4a). To mimic physiological conditions, his$_6$-tagged JAM-A molecules were immobilized in a physiologically oriented manner onto an NTA-Ni$^{2+}$-coated gold surface (Fig. 4a)[17], and surface chemistry was validated using AFM scratching experiments (see methods and Supplementary Fig. 2e). Specific binding forces were observed in the range of 20 to 130 pN and displayed on DFS plots for both the JAM-A-T3SA+ (Fig. 4b, upper panel) and JAM-A-T3SA− (Fig. 4b, lower panel) interactions. The JAM-A-reovirus interaction can be described by a single energy barrier with $x_u = 0.71 \pm 0.05$ nm and $k_{off} = 0.04 \pm 0.01$ s$^{-1}$ for the JAM-A-T3SA+ bond and $x_u = 0.48 \pm 0.03$ nm and $k_{off} = 0.05 \pm 0.03$ s$^{-1}$ for the JAM-A-T3SA− bond. While the off-rates are comparable, the distance to the

transition state is smaller for T3SA−, indicating that the energy landscape is described by a narrower energy valley that can accommodate less conformational variability. For T3SA− binding to JAM-A, we frequently observed larger binding forces, which corresponds to multiple interactions. Together with the narrower energy valley, this observation suggests that the single point mutation in T3SA− σ1 leads to a more rigid or compact conformation of the protein. Injection of a JAM-A antibody (Ab) reduced the binding frequency, confirming the specificity of virion-JAM-A binding (Fig. 4c).

To define the interaction of reovirus with JAM-A under physiological conditions, we evaluated reovirus binding to Lec2 cells expressing JAM-A. Combined optical and FD-based AFM were conducted using living fluorescently labeled Lec2 cells (nuclear eGFP and actin-mCherry) co-cultured with unlabeled Lec2-JAM-A cells (Fig. 5a–d). Adhesion maps (Fig. 5d) recorded with an AFM tip functionalized with T3SA+ virions revealed a higher density of adhesion events on Lec2-JAM-A cells (~3.5%, Fig. 5d, f), with rupture forces ranging from 50 to 400 pN. In contrast, Lec2 cells rarely displayed binding events (<0.8%, Fig. 5d, f), confirming the specificity of the interaction between JAM-A receptors expressed at the cell surface and T3SA+ virions (see also consecutive mapping in Supplementary Fig. 5a–d). To eliminate the contribution of the minimal SA expression on Lec2 cells, we also probed the interaction between T3SA− and Lec2-JAM-A cells and observed a similar frequency (~4.0%, Fig. 5f). In addition, the specificity of the interaction was verified using (i) JAM-A Ab (Supplementary Fig. 5k–n, Fig. 5f) and (ii) flow cytometry (Fig. 5g). Similar to results obtained using the in vitro approach, alteration of the SA-binding site does not influence reovirus binding to JAM-A (Supplementary Fig. 5e–h). Together, these results reveal that T3SA+ establishes stable weak (low multivalency) interactions with JAM-A independent of SA engagement.

To better define the function of JAM-A as a specific reovirus receptor on living cells, we analyzed the JAM-A binding forces and overlaid the data onto the DFS plot obtained previously using the model surface (Fig. 5e). Compared with the data acquired in vitro, we observed using living cells, establishment of up to four simultaneous uncorrelated virus-receptor bonds (WE-model, dashed curves). Similarly, binding forces measured with T3SA− virions on Lec2-JAM-A cells were extracted and overlaid onto the

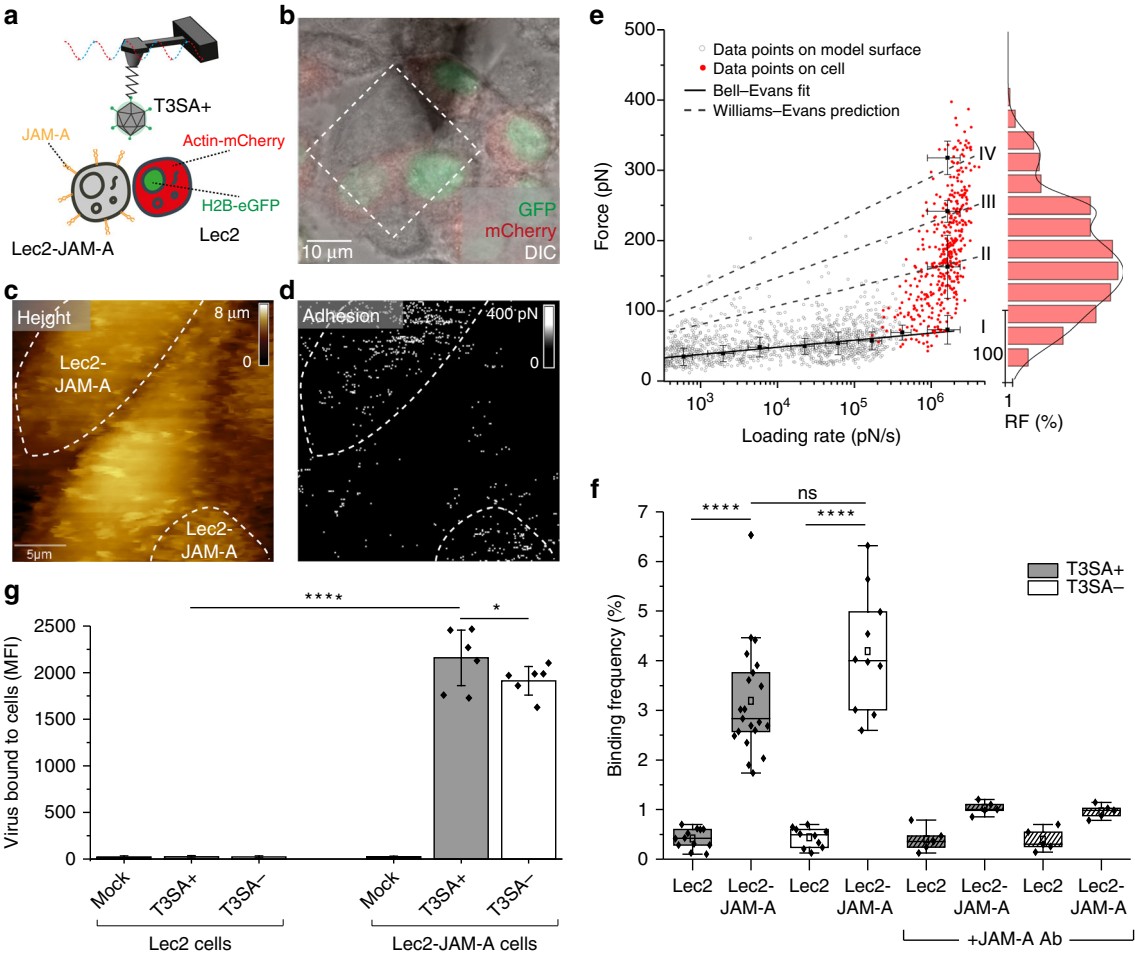

**Fig. 5** Probing reovirus binding to JAM-A on living cells. **a** Combined optical microscopy and FD-based AFM of T3SA+ interaction with JAM-A on living Lec2 cells. **b** Overlay of DIC, eGFP, and mCherry signals of a confluent layer of co-cultured fluorescent Lec2 cells (actin-mCherry and H2B-eGFP) and unlabeled Lec2-JAM-A cells. **c**, **d** FD-based AFM topography image (**c**) and corresponding adhesion map (**d**) of adjacent cells indicated in the dashed square in **b**. The adhesion map shows interactions mainly between T3SA+ particles and Lec2-JAM-A cells (white pixels). For higher visibility, the pixel size in the adhesion image was enlarged two-fold. **e** DFS plot of T3SA+ interactions with JAM-A on model surfaces (grey circles, taken from Fig. 4b—upper panel) and living cells (red dots). Histogram of the force distribution observed on cells fitted with a multi-peak Gaussian distribution ($n = 600$ data points) is shown on the side. **f** Box plot of BF observed for T3SA+ (grey) and T3SA− (white) virions, with (hatched lines) and without injection of JAM-A Ab (10 μg/ml). The horizontal line within the box indicates the median, boundaries of the box indicate the 25th and 75th percentile, and the whiskers indicate the highest and lowest values of the results. The square in the box indicates the mean. For all experiments, data are representative of at least $n = 5$ independent experiments. **g** Influence of JAM-A on virus binding determined using flow cytometry. Cells were incubated with either no virions (Mock) or Alexa Flour 488-labeled T3SA+ or T3SA− virions ($10^5$ particles per cell), and the median fluorescence intensity (MFI) of cell-bound virus was determined by flow cytometry as shown in Supplementary Fig. 3a. Error bars indicate s.d. of the mean value. Experiments were repeated twice ($n = 2$ independent experiments, each with duplicate samples). ns, $P > 0.05$; *$P < 0.05$; ****$P < 0.0001$; determined by two-sample $t$-test in Origin (**f**) and by two-way ANOVA corrected for multiple comparisons using Tukey's test in GraphPad Prism (**g**), respectively. Source data are provided as a Source Data file

DFS plot (Supplementary Fig. 5i, j) providing similar results in terms of binding frequency (Fig. 5f) and number of simultaneous uncorrelated virus-receptor bonds established on living cells (Supplementary Fig. 5j). However, for both T3SA+ and T3SA−, the majority of adhesion events show rupture forces corresponding to rupture of one or two JAM-A receptor interactions (Fig. 5e, Supplementary Fig. 5j, histogram). These findings suggest that binding to JAM-A is kinetically or sterically less favored than binding to α-SA on cells, for which up to six bonds were observed under similar experimental conditions.

**α-sialylated glycans trigger multivalent reovirus binding**. As both α-SA and JAM-A act in concert during reovirus attachment to the cell surface, we also assessed reovirus binding to JAM-A in the presence of α-SA, first using model surfaces (Fig. 6 and

Supplementary Fig. 6) and then using living cells (Fig. 7). While probing T3SA+ binding to JAM-A in vitro, we injected 1 mM glycans with (Neu5Ac and LSTa) and without (LNnT) terminal α-SA. Remarkably, addition of both Neu5Ac and LSTa (Fig. 6b, c) led to a strong increase in the overall binding force (up to 400 pN, compared to 130 pN in absence of α-SA (Fig. 4b)), indicating a shift towards strong multivalent interactions. In the presence of α-SA, three or more simultaneous uncorrelated virus-receptor bonds were observed between T3SA+ and JAM-A (Fig. 6b, c), giving rise to an increase in the overall avidity. In contrast, incubation with LNnT had no effect on the binding of T3SA+ to JAM-A (Fig. 6d), suggesting that the sialyl group is required to promote this property. This behavior was not observed for T3SA− (Supplementary Fig. 6a–c), suggesting that triggering multivalent interactions by α-SA can be attributed to the formation of

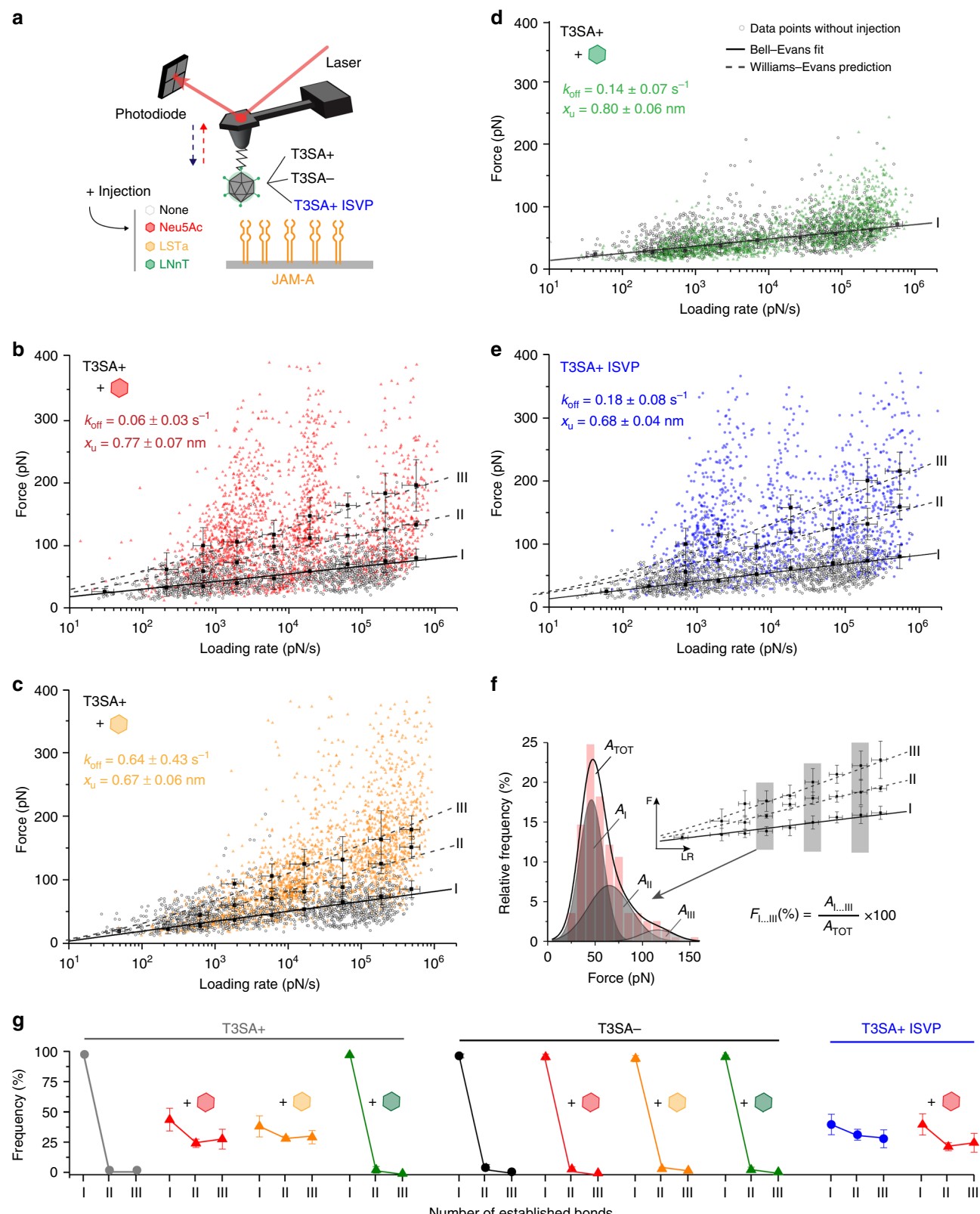

a complex between the sialylated glycan and the glycan-binding site in the serotype 3 σ1 tail domain (Fig. 1b).

Generally, processes used by viruses to enter cells are characterized by a complex series of events initiated by the binding of viral surface proteins to cellular receptors, often inducing conformational changes in viral capsid components. We wondered whether the binding of α-SA to σ1 could induce such a conformational change, favoring, for example, a more extended conformation. To test this hypothesis, we probed the binding potential of T3SA+ infectious subvirion particles (ISVPs) (Fig. 6e). After limited proteolytic treatment of virions to generate ISVPs, cryoEM images reveal σ1 in a more extended state, projecting radially away from the particle surface[27] (Fig. 1a). This observation provides the rationale to test whether

**Fig. 6** Influence of sialylated glycans on reovirus binding to JAM-A. **a** Binding of T3SA+ or T3SA− virions, or T3SA+ ISVPs to JAM-A was monitored following injection of 1 mM α-SA glycans (Neu5Ac [**b**, red] and LSTa [**c**, yellow]) or non-sialylated glycan (LNnT [**d**, green]). **b**–**d** DFS plots of interaction forces measured between T3SA+ and JAM-A after adding 1 mM glycan (Neu5Ac in **b**, LSTa in **c**, and LNnT in **d**). Grey dots represent the measured binding forces before injection. **e** DFS plot of the interaction forces between JAM-A and T3SA+ ISVPs, which display a more extended conformation of the σ1 protein. Multivalent interactions are observed for T3SA+ ISVPs (blue) in comparison to T3SA+ virions (gray) without injection of free SA. **f** Example for analysis of the number of established bonds shown in panel **g**. Relative frequency of single and multiple bonds before and after adding free glycans was determined from areas under respective peaks ($A_I$, $A_{II}$, $A_{III}$; grey shaded) within force distribution histograms (red bars with cumulative peak, $A_{TOT}$). **g** Number of bonds established between JAM-A and T3SA+ (left panel) or T3SA− (middle panel) virions or T3SA+ ISVPs (right panel), before and after injection of sialylated (Neu5Ac—red, LSTa—yellow) or non-sialylated (LNnT—green) glycans. Error bars indicate s.d. of the mean value. For all experiments, data are representative of at least $n = 3$ independent experiments. Source data are provided as a Source Data file

ISVP-JAM-A interactions mimic virion-JAM-A interactions in the presence of α-SA, which may induce an extended conformer of σ1. Remarkably, we observed strong interactions between T3SA+ ISVPs and JAM-A, which were comparable to the interactions of T3SA+ virions with JAM-A following incubation with α-SA, suggesting that α-SA binding to σ1 induces a conformational change in the protein that enhances its affinity for JAM-A. In terms of number of bonds established between reovirus virions and JAM-A, our analysis reveals that after binding to α-SA, the reovirus-JAM-A binding potential is increased to a level similar to that of ISVP-JAM-A interaction (Fig. 6f, g). We also tested whether binding of ISVPs to JAM-A could be increased by treatment with free α-SA (Fig. 6g, Supplementary Fig. 6d, e). However, there was no observable change in ISVP-JAM-A interactions following α-SA treatment. Therefore, after activation of T3SA+ virions by α-SA, the σ1 protein appears to undergo a conformational change, perhaps to a more extended form like that in ISVPs.

To test whether this activation mechanism occurs in a cellular context, we probed reovirus binding to Lec2-JAM-A cells and monitored the adhesion behavior following injection of Neu5Ac, LSTa, or LNnT (Fig. 7, Supplementary Fig. 7). As Lec2-JAM-A cells are deficient in sialylated glycan expression and the binding of reovirus to Lec2 cells is rare (<1%; see Supplementary Fig. 7), we hypothesized that most interactions of reovirus with Lec2-JAM-A cells are established via JAM-A receptors. Therefore, we used this cell line to study the effect of injected glycan derivatives on reovirus-JAM-A interactions and compared the overall binding frequency before and after glycan injection. T3SA+ virions displayed a significant increase in binding of ~20–25% following injection of α-SA-glycans (Neu5Ac and LSTa) (Supplementary Fig. 7; from 3.9 to 4.9% for Neu5Ac and from 3.8 to 4.8% for LSTa). In contrast, we did not observe an increase in binding after injection of LNnT, which lacks α-SA (Supplementary Fig. 7; from 3.8 to 3.9%). To evaluate whether the residual SA on Lec2 cells could influence our results, we treated Lec2 cells with neuraminidase (40 mUnit/mL for 1 h) to cleave residual cell-surface α-SA-glycans (Supplementary Fig. 8). Similar to the results gathered using untreated cells, we observed an increase in T3SA+ binding of ~20% after injection of Neu5Ac on neuraminidase-treated Lec2 cells. These results suggest that interaction with SA enhances reovirus binding to JAM-A on the cell surface, and residual SA on the Lec2 cell surface minimally contributes to interaction with reovirus.

To quantify the multivalence of reovirus binding to cell-surface receptors, we analyzed the binding force distribution (Fig. 7p–s). Examination of adhesion images showing binding events within the high-force range (Fig. 7c, e, h, j, m, o) indicated that the frequency of these events significantly increased after incubation with α-SA. This observation is consistent with our in vitro data confirming a change in the virion binding potential following α-SA incubation. Analysis of the number of bonds established between a T3SA+ virion and JAM-A on the cell surface (Fig. 7t)

confirmed this enhanced multivalence in the presence of α-SA. Together, these data indicate that α-SA binding to σ1 increases the affinity of σ1 for JAM-A as well as the avidity of the virus for the cell surface, likely as a consequence of an α-SA-induced conformational change.

**Multivalent anchorage alters binding and diffusion potential.** To evaluate how α-SA influences the dynamics of reovirus binding to JAM-A, we used optical biolayer interferometry (BLI) and fluorescence microscopy-based single-particle tracking (SPT). Using BLI, we quantified reovirus binding to $Ni^{2+}$-NTA-biosensors coated with JAM-A and also tested the influence of free Neu5Ac on the overall avidity. The data show that T3SA+ and T3SA− bind to JAM-A with high avidity ($K_D$ ~nM range) (Fig. 8a). As expected, free Neu5Ac had no influence on T3SA− binding. In marked contrast, T3SA+ virions incubated with free Neu5Ac and ISVPs have a much higher avidity for JAM-A, reaching a very high affinity ($K_D$ ~pM range) (Fig. 8a). These observations are consistent with our AFM data showing a binding potential of T3SA+ virions incubated with α-SA compounds comparable to that of ISVPs.

The dynamics of reovirus binding to the surface of living cells was evaluated by SPT using high-speed confocal microscopy. Fluorescently labeled T3SA+ virions were incubated with mCherry-labeled Lec2-JAM-A cells co-cultured with CHO-JAM-A cells. Time-lapse series of images were recorded in the presence or absence of 1 mM Neu5Ac (Fig. 8b–e). Analysis of at least 15 particle trajectories on each cell type revealed that virions diffuse over greater distances and with increased speeds on cells lacking α-SA glycans (Fig. 8f). Moreover, injection of free α-SA reduced T3SA+ diffusion on the cell surface, presumably by the capacity to mediate multivalent interactions with JAM-A, and significantly increased the number of particles bound to CHO-JAM-A cells (Fig. 8f). Importantly, this effect was not observed for the non-SA-binding strain T3SA− (Fig. 8f, Supplementary Fig. 9). Together, these observations strengthen our conclusion that σ1 binding to α-SA induces a conformational change in σ1 that leads to an increase in the multivalent attachment of the virus to cell-surface receptors.

**Discussion**
In this study, we used AFM in combination with confocal microscopy to force-probe and define interactions between essential components of the cellular plasma membrane and reovirus that facilitate virus-cell entry. The crystal structure of reovirus attachment protein σ1 reveals an elongated fiber with a tail domain formed by α-helical coiled coil and triple-β spiral[28] and a head domain formed by a compact eight-stranded β-barrel[29]. As the principal factor initiating reovirus cell entry, σ1 of serotype3 reovirus contains receptor-binding regions in both tail and head domains. While the triple-β spiral in the tail domain binds to α-SA[20], the head domain binds to JAM-A[30]. Our current

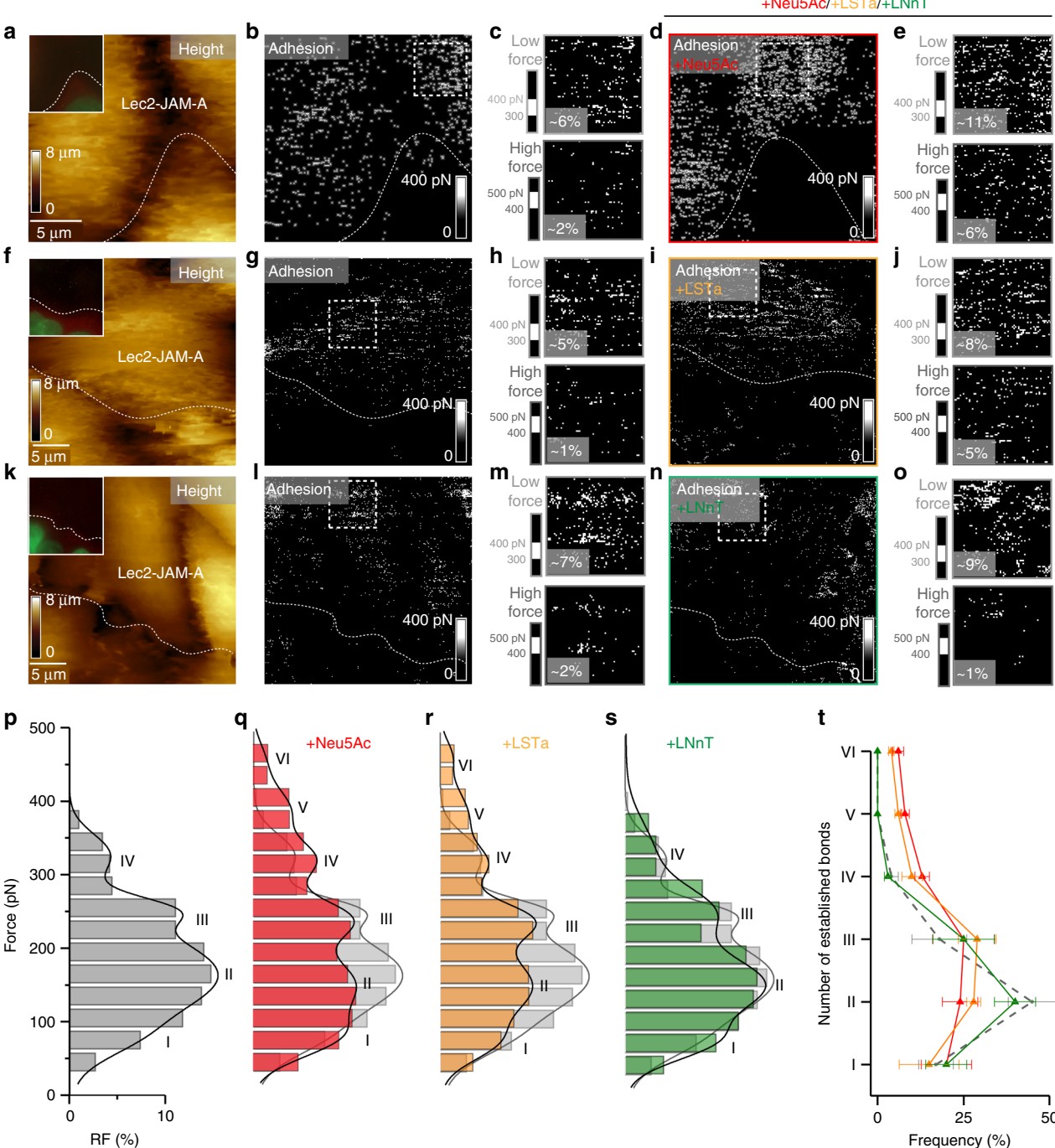

**Fig. 7** Monitoring the effect of SA addition on reovirus binding to living cells. T3SA+ binding to Lec2-JAM-A cells was assessed before and after adding 1 mM Neu5Ac (**a–e**), 1 mM LSTa (**f–j**), or 1 mM LNnT (**k–o**). **a, f, k** AFM topography image of adjacent Lec2 and Lec2-JAM-A cells with fluorescent image (20 × 20 μm) inset showing fluorescently-tagged Lec2 cell lacking JAM-A expression. **b, g, l** Corresponding adhesion map recorded before injection of glycan. **c, h, m** Enlarged images of adhesion maps recorded on Lec2-JAM-A cells (dashed square in adhesion map). The upper images display the lower force range (300 to 400 pN), whereas the lower images display the higher force range (400 to 500 pN), with significantly fewer adhesion events. **d, i, n** Adhesion maps recorded following injection of free Neu5Ac (**d**), LSTa (**i**), or LNnT (**n**). The area probed is similar to the area recorded before glycan injection. **e, j, o** Enlarged images of adhesion maps recorded on Lec2-JAM-A cells (dashed square in adhesion map and similar areas as in **b, g, i** show more adhesion events in the high-force range upon injection of sialylated glycan [Neu5a and LSTa] and no significant change for non-sialylated glycan [LNnT]). The frequency of adhesion events is indicated. (**p–s**) Histogram of the force distribution extracted on Lec2-JAM-A cells (dashed square in adhesion map) before (**p**) and after adding Neu5Ac (**q**), LSTa (**r**), and LNnT (**s**) (n > 700 for each condition). Histograms were fitted with a multi-peak Gaussian distribution. **t** Number of bonds established between JAM-A cell-surface receptors and T3SA+ before (grey dashed) and after injection of sialylated or non-sialylated glycans (colored). Error bars indicate s.d. of the mean value. The statistical analysis is shown in Supplementary Table 1. For all experiments, data are representative of at least n = 15 cells from n = 5 independent experiments. Source data are provided as a Source Data file

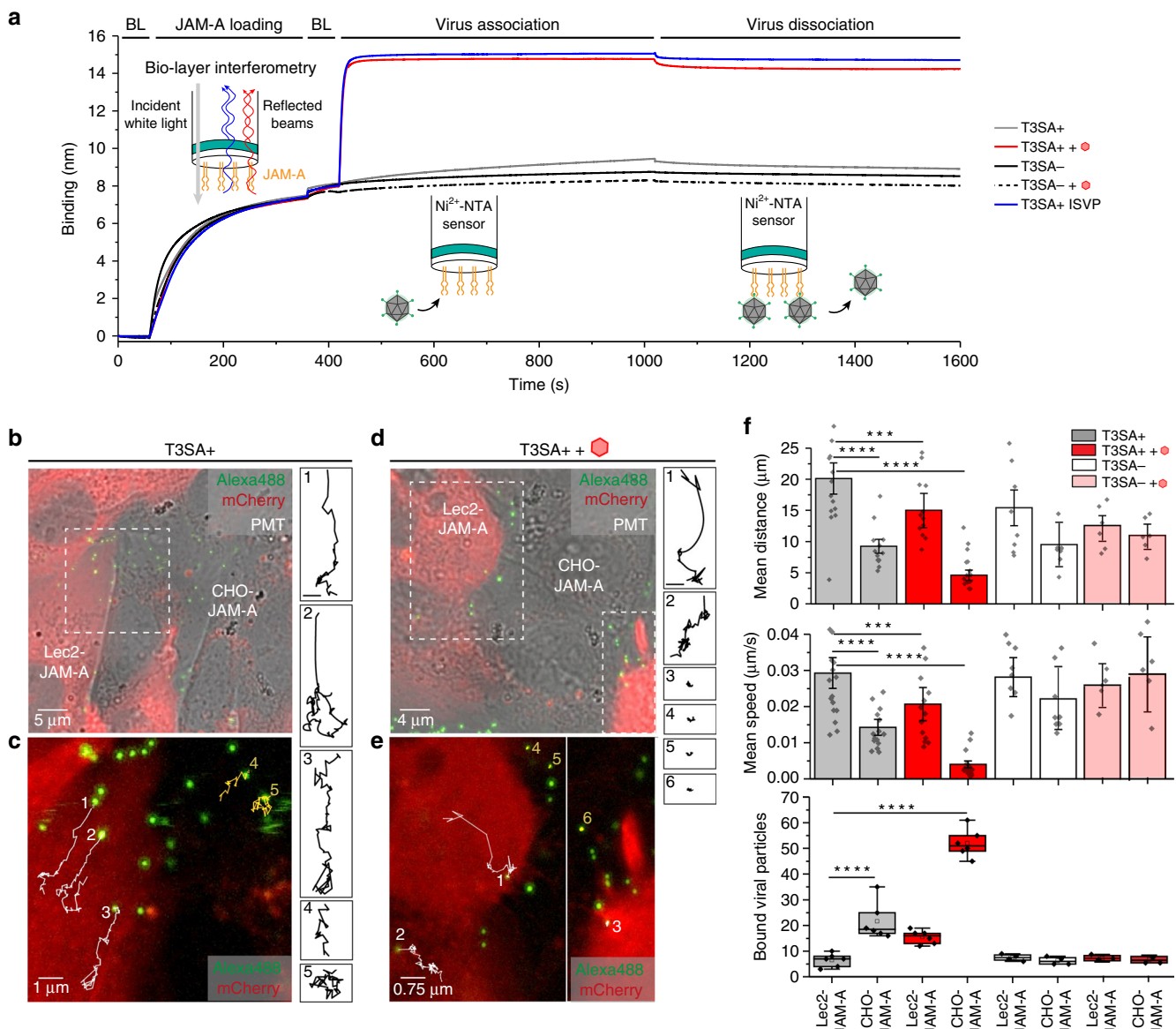

**Fig. 8** Multivalent anchorage of virions alters diffusion potential and binding behavior. **a** Biolayer interferometry data for the binding of reovirus (T3SA−, T3SA+ and T3SA+ ISVP) to JAM-A receptor immobilized on NTA-coated biosensors. The effect of addition of 1 mM Neu5Ac in solution was tested for both T3SA− and T3SA+. Sensorgram starts with baseline (BL) measurement following by the immobilization of JAM-A to the NTA biosensor (loading), the addition of the virions (association), and finally by the dissociation phase. **b–f** Real-time confocal fluorescence imaging of reovirus particles (labeled with Alexa Fluor 488 dye) incubated with co-cultured CHO-JAM-A and fluorescently labeled Lec2-JAM-A cells in the absence (**b**, **c**) and presence (**d**, **e**) of 1 mM Neu5Ac. **b**, **d** Overlay images of Alexa Fluor 488 (virions), mCherry-actin (Lec2-Jam-A), and PMT signals. **c**, **e** Time-lapse trajectories of T3SA+ particles. White and yellow trajectories represent the movement on Lec2-JAM-A cells and CHO-JAM-A cells, respectively. Magnification of each trajectory is shown on the right side with the corresponding number (scale bar: 1 µm). **f** Analysis of the mean travelled distance (top panel), mean travel speed (middle panel), and bound viral particles (bottom panel) for T3SA+ binding in the absence (grey) or presence (red) of Neu5Ac as well as for T3SA− binding in the absence (white) or presence (light red) of Neu5Ac following adsorption to the cell mixture. The horizontal line within the box plot (bottom panel) indicates the median, boundaries of the box indicate the 25th and 75th percentile, and the whiskers indicate the highest and lowest values of the results. The square in the box indicates the mean. Data are representative of at least n = 3 independent experiments, with a minimum of n = 15 analyzed trajectories each. ***P < 0.001; ****P < 0.0001; determined by two-sample t-test in Origin. Error bars indicate s.d. of the mean value. Source data are provided as a Source Data file

understanding of reovirus attachment and cell entry comes largely from either in vitro assays such as surface plasmon resonance[15] and solid-phase binding studies or experiments using fixed cells such as analysis of binding and internalization of fluorescent virions[31]. The in vitro assays are ensemble methods that provide information only about average properties, and the fluorescence assays using fixed cells provide limited quantitative

information about the dynamics of the virus-cell interactions during critical early steps in infection.

Using single-virus force spectroscopy, we investigated reovirus binding to both α-SA and JAM-A by quantifying the binding strength to each receptor and extracting the kinetic parameters characterizing single bonds. AFM experiments conducted using model surfaces and living cells enabled us to determine the

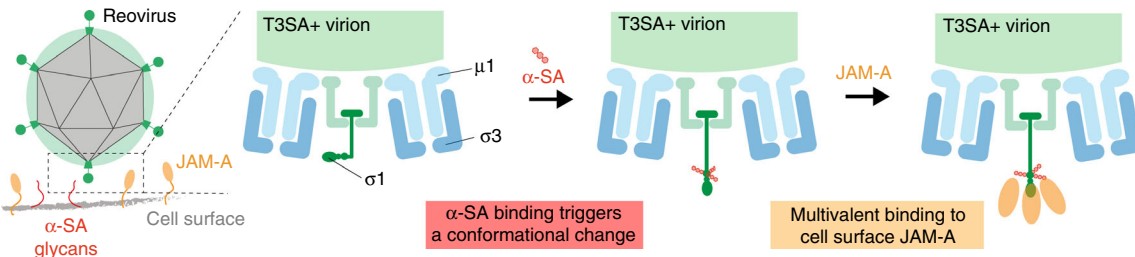

**Fig. 9** Glycan-mediated enhancement of reovirus receptor binding. Upon binding of α-SA, the σ1 outer capsid proteins undergo a conformational change leading to a more extended conformation. This results in an increased affinity for JAM-A

valency of interactions. During reovirus binding to the cell surface, we observed that three parallel interactions with α-SA and two to three interactions with JAM-A are favored. This finding suggests that receptor-binding domains on each monomer of the σ1 homotrimer can independently engage receptors, α-SA and JAM-A. These results also establish that the number of bonds contributes to the overall avidity of virus-receptor binding. Thus, the affinity for an individual receptor molecule might be very low (in the mM range for single protein-glycan interactions)[32] but can increase to remarkable avidity values (in the nM range) as a consequence of multivalent interactions[33,34]. In the case of reovirus, after landing on the cell surface and binding to receptors, the virus adheres to a confined location from which it will be endocytosed in a signal-induced manner[3]. Maximizing the number of bonds may help reduce the lateral diffusion of virions. In this context, extraction of the number of virus-receptor bonds at a single-virion level, as conducted in this study, will allow us to understand the initiation of the infection process.

There are few examples of cooperativity between attachment factors and specific entry receptors. Attachment factors often mediate weak interactions that lack specificity and serve to tether the virus to the cell surface allowing the virus to access specific entry mediators. We discovered that the attachment of the reovirus σ1 tail to α-SA enhances the binding of the σ1 head to JAM-A. Since the α-SA-mediated increased affinity of virions for JAM-A mimics the JAM-A affinity of ISVPs, which contain a more extended conformer of σ1[27], our data suggest that σ1 binding to α-SA triggers a conformational change in the protein that renders the JAM-A-binding site more accessible. From a molecular point of view, the structure of σ1 in complex with sialylated oligosaccharides reveals a *trans* configuration of the Leu203-Pro204 peptide bond[20]. While peptide bonds are nearly always found in the *trans* configuration, *cis* configurations are sometimes observed with peptidyl-prolyl bonds[35]. For rotavirus, the structure of receptor-binding protein VP4 in complex with α-SA was determined at 100 K and room temperature (295 K). The Gly156-Pro157 peptide bond adjacent to the SA-binding site is predominantly in the *trans* configuration at room temperature, whereas *cis-trans* isomerization was more strongly evident at 100 K[36]. Therefore, an attractive hypothesis is that α-SA binding to the σ1 tail induces a *cis* to *trans* isomerization of the L203-P204 bond resulting in an important conformational change towards a more extended form of the protein (Fig. 9).

Findings reported here elucidate the complex interplay between reovirus and its cellular receptors prior to viral entry. Binding to α-SA, which is engaged with low affinity, serves as the initial attachment event and triggers a conformational change that enhances further specific interactions with the high-affinity JAM-A receptor. This two-step adhesion-strengthening mechanism provides evidence for glycan-mediated cell targeting. Moreover, our findings provide unique opportunities to manipulate reovirus binding efficiency and infectivity for vaccine and oncolytic applications.

## Methods

**Generation of reovirus stocks**. Stocks of reovirus strains T3SA+ and T3SA− were prepared by plaque purification and passaging the viruses 3–4 times in L929 cells (ATCC, #CCL-1). Infected cells were lysed by sonication, and virions were extracted from lysates using vertrel-XF[37,38]. The extracted virions were layered onto 1.2 to 1.4 g/cm³ caesium chloride step gradients and centrifuged at 25000 rpm at 4 °C for 18 h. The band corresponding to the density of reovirus particles (~1.36 g/cm³)[39] was collected and exhaustively dialyzed against virion-storage buffer (150 mM NaCl, 15 mM MgCl₂, and 10 mM Tris [pH 7.4]). Particle concentration was determined from optical density at 260 nm (1 OD$_{260}$ = 2.1 × 10$^{12}$ particles mL$^{-1}$)[39]. Viral titers were determined by plaque assay using L929 cells[40].

ISVPs were prepared by digesting virions (2 × 10$^{12}$ particles/mL) with 2 mg/mL α-chymotrypsin (Sigma–Aldrich) at 37 °C for 60 min[41]. The reaction was quenched by incubation on ice and addition of phenylmethylsulfonyl fluoride (Sigma–Aldrich) to a concentration of 2 mM.

For fluorescent labeling, reovirus particles were diluted into fresh 50 mM sodium bicarbonate (pH 8.5; 6 × 10$^{12}$ particles/mL) and incubated with 20 μM succinimidyl ester of Alexa Flour 488 (Invitrogen) at room temperature for 90 min in the dark[42]. Unreacted dye was removed by dialysis against PBS at 4 °C overnight.

**Engineering and characterization of JAM-A expressing cells**. Monolayers of CHO (ATCC, #CCL-61) and Lec2 (ATCC, #CRL-1736) cells were transduced with lentiviruses encoding a puromycin-resistance gene and human JAM-A or a puromycin-resistance gene alone. Transduced cells were selected for puromycin resistance by passaging twice in medium containing 20 μg mL$^{-1}$ puromycin. The concentration of puromycin used was the minimal concentration that yielded complete death of non-transduced CHO and Lec2 cells. Following selection for puromycin resistance, cells were further selected for cell-surface expression of JAM-A using fluorescence-activated cell sorting (FACS). Cell-surface expression of JAM-A was detected using the monoclonal antibody, J10.4 (provided by Charles Parkos, Emory University; used at 1:1000 in flow cytometry)[43], and a fraction of cells with high JAM-A expression was collected and propagated using puromycin selection. In this manuscript, cells transduced and selected for puromycin resistance alone will be referred to as CHO and Lec2 and those selected for both puromycin resistance and JAM-A expression will be referred to as CHO-JAM-A and Lec2-JAM-A.

**Culture of cell lines**. CHO cells (CHO, CHO-JAM-A) were grown in Ham's F12 medium (Sigma–Aldrich) supplemented to contain 10% fetal bovine serum (FBS), penicillin (100 U mL$^{-1}$), and streptomycin (100 μg mL$^{-1}$) (Invitrogen) at 37 °C in a humidified atmosphere with 5% CO₂. During alternate passages, 20 μg mL$^{-1}$ puromycin was added to the medium. Lec2 cells (Lec2, Lec2-JAM-A) were grown in Mem α, nucleosides medium (Gibco) supplemented to contain 10% FBS, penicillin (100 U mL$^{-1}$), and streptomycin (100 μg mL$^{-1}$) at 37 °C in a humidified atmosphere with 5% CO₂. During alternate passages, 20 μg mL$^{-1}$ puromycin was included in the medium.

**Transduction of CHO and Lec2 cells**. The four cell types used in this study were transduced to express nuclear eGFP as well as cytoplasmic mCherry using H2B-eGFP and actin-mCherry-expressing lentiviruses, respectively[44]. Cells expressing both eGFP and mCherry were selected by FACS and propagated using the culture conditions described above. For single-particle tracking experiments, Lec2-JAM-A expressing mCherry alone were selected and propagated as described above.

**FACS of transduced cells**. Cells transduced with H2B-eGFP and actin-mCherry transgenes were trypsinized and collected into phosphate buffered saline (PBS) containing 2 mM EDTA and 1% FBS. Cells were sorted using a BD FACSARIA III cell sorter, with a nozzle of 85 μm, sheath pressure of 45 psi, drop frequency of 47 kHz, and sort precision of 0-32-0. eGFP was excited with a 488 nm laser and emission was filtered with a 530/30 band-pass filter and a 505 long-pass mirror. mCherry was excited with a 561 nm (yellow-green) laser and emission was filtered with a 610/20 band-pass filter. Cells expressing both eGFP and mCherry were collected and propagated using the culture conditions described above.

**Functionalization of AFM tips**. NHS-PEG$_{27}$-acetal linkers were used to functionalize AFM tips as described[45]. AFM tips (PFQNM-LC and MSCT probes, Bruker) were immersed in chloroform for 10 min, rinsed with ethanol, dried with a stream of filtered nitrogen, cleaned for 10 min using an ultraviolet radiation and ozone (UV-O) cleaner (Jetlight), and immersed overnight in an ethanolamine solution (3.3 g of ethanolamine hydrochloride in 6.6 mL of DMSO). The cantilevers were washed three times with DMSO and two times with ethanol and dried with nitrogen. To ensure a low grafting density of the linker on the AFM tip, 1 mg of acetal-PEG$_{27}$-NHS was diluted in 0.5 mL of chloroform with 30 μL of triethylamine[45]. Ethanolamine-coated cantilevers were immersed for 2 h in this solution, washed three times with chloroform, and dried with nitrogen. Cantilevers were then immersed for 10 min in 1% citric acid in milliQ water, washed three times with milliQ water, and dried with nitrogen. Virus solution (80 μL at ~10$^8$ to 10$^9$ particles mL$^{-1}$) was pipetted onto the tips placed on Parafilm (Bemis NA) in a small plastic dish stored within an icebox. A freshly prepared solution of NaCNBH$_3$ (2 μL at ~6% wt. vol$^{-1}$ in 0.1 M NaOH$_{(aq)}$) was gently mixed into the virus solution, and the cantilever chips were gently positioned with the cantilevers extending into the virus drop. The icebox was incubated at 4 °C for 1 h. Then, 5 μL of 1 M ethanolamine solution (pH 8) was gently mixed into the drop to quench the reaction. The icebox was incubated at 4 °C for an additional 10 min, and the cantilever chips were removed, washed three times in ice-cold PBS, and stored in individual wells of a multiwell dish containing 2 mL of ice-cold virus buffer (150 mM NaCl, 15 mM MgCl$_2$, 10 mM Tris, pH adjusted to 7.4) per well until used in AFM experiments. During these functionalization steps, the virus-functionalized cantilevers were never allowed to dry. Transfer of the functionalized AFM cantilevers to virus buffer and then to AFM was rapid (≤20 s) and, during transfer, a drop of virus buffer remained on the cantilever and tip. Cantilevers were used in AFM experiments the same day they were functionalized. Control experiments using confocal imaging showed that in most cases no more than one viral particle was present at the apex of the AFM tip, which interacts with a model surface or cell surface during an AFM experiment[18].

**Preparation of α-SA-coated model surfaces**. Biotinylated α2,3-linked SA was immobilized on plates using the biotin-streptavidin system[16] as described[17]. Gold-coated silicon substrates were incubated at 4 °C overnight in a 25 μg mL$^{-1}$ solution of biotinylated bovine serum albumin (BBSA, Sigma–Aldrich) in PBS. After rinsing with PBS, the BBSA surfaces were exposed to a 10 μg mL$^{-1}$ solution of streptavidin (Sigma–Aldrich) in PBS for 2 h, following by rinsing with PBS. The BBSA-streptavidin surfaces were immersed for 2 h in a 10 μg mL$^{-1}$ solution of biotinylated 3'-sialyl-N-acetyllactosamine (α2,3-linked SA, Dextra) in PBS, followed by rinsing with PBS. The surfaces showed a homogeneous and stable morphology under repeated scanning and displayed a thickness of ~2 nm. The thickness of the deposited layer was estimated by scanning a small area (0.5 × 0.5 μm$^2$) of the surface at high forces to remove the attached biomolecules, followed by imaging larger squares of the same region (2.5 × 2.5 μm$^2$) at a lower force.

**Preparation of JAM-A-coated model surfaces**. His$_6$-tagged JAM-A (Bio-Connect Life Science) was immobilized using NTA-His$_6$ binding chemistry. Gold-coated surfaces were rinsed with ethanol, dried with a gentle nitrogen flow, cleaned for 15 min by UV and ozone treatment, and immersed overnight in ethanol containing 0.05 mM of NTA-terminated (10%) and triethylene glycol(EG)-terminated (90%) alkanethiols. After rinsing with ethanol, the samples coated with alkanethiols were immersed in a 40 mM aqueous solution of NiSO$_4$ (pH 7.2) for 1 h, rinsed with water, incubated with his$_6$-tagged JAM-A (0.1 mg mL$^{-1}$) for 2 h, and rinsed with PBS. The functionalized surfaces were kept hydrated and used immediately after preparation. The surfaces showed a homogeneous and stable morphology under repeated scanning and displayed a thickness of ~3 nm. The thickness was measured as described for sialic-acid-coated model surfaces.

**FD-based AFM on model surfaces**. AFM Nanoscope Multimode 8 (Bruker) was used (Nanoscope software v9.1) to conduct FD-based AFM. Virus-functionalized MSCT-D probes (with spring constants calculated using thermal tune[46], ranging from 0.024 to 0.043 N m$^{-1}$) were used to record force curves from 5 × 5 μm arrays in the force-volume (contact) mode. The approach velocity was kept constant at 1 μm s$^{-1}$, and retraction velocities were varied from 0.1, 0.2, 1, 5, 10 to 20 μm s$^{-1}$ to ensure that the energy landscape between the virus and its cognate receptor was probed over a wide range of loading rates. The pulling velocity (v) and loading rate (LR) can be related as follows:

$$ LR = \Delta F / \Delta t = k_{eff} \cdot v \qquad (1) $$

where ΔF/Δt being the applied force over time, and k$_{eff}$ the effective spring constant of the system. The ramp size was set to 500 nm and the maximum force to 500 pN, with no surface delay. The sample was scanned using a line frequency of 1 Hz, and 32 pixels were scanned per line (32 lines in total with 1024 data points [FD curves] per retraction speed). All FD-based AFM measurements were obtained in virus buffer at ~25 °C. Force curves were analyzed using the Nanoscope analysis software v1.7 (Bruker). To identify peaks corresponding to adhesion events occurring between particles linked to the PEG spacer and the receptor model surface, the retraction curve before bond rupture was fitted with the worm-like chain model for

polymer extension[47]. The latter expresses the force-extension (F-x) relationship for semi-flexible polymers and is described by the following equation, with l$_P$ the persistence length, L$_c$ the contour length and k$_b$T the thermal energy:

$$ F = \frac{k_b T}{l_P} \left( \frac{1}{4 \left(1 - \frac{x}{L_c}\right)^2} + \frac{x}{L_c} - 0.25 \right) \qquad (2) $$

Origin software (OriginLab) was used to display the results in dynamic force spectroscopy (DFS) plots to fit histograms of rupture force distributions for distinct loading rate ranges and to apply various force spectroscopy models as described[18,19,48].

**FD-based AFM and fluorescence microscopy on living cells**. Correlative images were acquired using an AFM (Bioscope Catalyst and Bioscope Resolve, Bruker) operated in the PeakForce QNM mode (Nanoscope software v9.2) to conduct FD-based AFM and coupled to an inverted epifluorescence microscope (Zeiss Observer Z.1) as described[19,49]. A 40x oil objective (NA = 0.95) was used. The AFM was equipped with a 150 μm piezoelectric scanner and a cell-culture chamber allowing to control the temperature, the humidity and the CO$_2$ concentration[18]. Overview images of cell surfaces (20–30 μm$^2$) were recorded at imaging forces of ~500 pN using PFQNM-LC probes (Bruker) having tip lengths of 17 μm, tip radii of 65 nm, and opening angles of 15°. All fluorescence microscopy and FD-based AFM imaging experiments were conducted under cell-culture conditions using the combined AFM and fluorescence microscopy chamber (Supplementary Fig. 1a) at 37 °C in either Mem α, nucleosides or Ham's F12 culture medium, depending on the cell type. A gas mixture of synthetic air with 5% CO$_2$ at 95% relative humidity using a gas humidifier membrane (PermSelect silicone) was infused at 0.1 L min$^{-1}$ into the microscopy chamber. The humidity was controlled using a humidity sensor (Sensirion). Cantilevers were first calibrated using the thermal noise method[50], yielding values ranging from 0.095 to 0.135 N m$^{-1}$ for PFQNM-LC probes. The AFM tip was oscillated in a sinusoidal fashion at 0.25 kHz with a 750 nm amplitude in the PeakForce Tapping mode. The sample was scanned using a frequency of 0.125 Hz and 256 pixels per line (256 lines). AFM images and FD curves were analyzed using the Nanoscope analysis software (v1.7, Bruker), Origin, and ImageJ (v1.52e). Individual FD curves detecting unbinding events between the virus and the cell surface were analyzed using the Nanoscope analysis and Origin software. The baseline of the retraction curve was corrected using a linear fit on the last 30% of the retraction curve. Using the force-time curve, the loading rate (slope) of each rupture event was determined (Supplementary Fig. 1c). Optical images were analyzed using Zen Blue software (Zeiss)[18,19,48].

**Monitoring the effect of SA addition**. The live cell experiments were conducted in the same manner as described above by scanning a suitable field of cells, followed by adding 1 mM of the respective glycan to the culture medium. The same area was scanned again to monitor potential changes after glycan addition. To assess specificity, blocking agents (1 mM Neu5Ac or 10 μg/ml JAM-A Ab [Sigma, # SAB4200468]) were added subsequently.

**Monitoring reovirus binding to neuraminidase-treated cells**. To remove residual cell-surface SA from Lec2 cells, the live cell experiments were conducted in the same manner as described above by scanning a suitable field of cells, followed by treatment with neuraminidase on the microscope stage to allow a second scan of the same field following treatment. The culture medium was removed, and cells were washed with 2 mL PBS (Sigma–Aldrich), treated with *Arthrobacter ureafaciens* neuraminidase (Sigma–Aldrich) at a final concentration of 40 mUnit/mL in PBS for 1 h, and washed with 2 mL PBS. Experiments were conducted using cell-culture medium without any supplements to suppress SA recovery. In addition, 1 mM Neu5Ac was added during a third scan and 10 μg/ml JAM-A Ab during a fourth scan to monitor SA-mediated changes and to assess the specificity of observed interactions, respectively.

**Quantification of reovirus and lectin binding**. CHO and Lec2 cells (Puro and JAM-A cell lines) were detached from cell-culture dishes using Cellstripper (Cellgro) at 37 °C for 15 min, quenched with the corresponding cell-culture medium, and washed once with PBS. To quantify reovirus binding, cells were adsorbed with 10$^5$ fluoresceinated reovirus virions per cell at 4 °C for 1 h. To compare cell-surface SA expression between cell lines, detached cells were adsorbed with fluorescein-labeled wheat germ agglutinin (WGA) (Vector laboratories, #FL-1021) at a concentration of 1 μg/mL in PBS containing 5% BSA at 4 °C for 1 h. After respective treatments, cells were washed twice with FACS buffer (PBS containing 2% FBS), stained with LIVE/DEAD Fixable Violet Dead Cell Stain kit (Invitrogen, #L34963) for 15 min, washed twice again with FACS buffer, and fixed in PBS containing 1% paraformaldehyde. Cells were analyzed using LSRII flow cytometer (BD Bioscience), and reovirus or lectin bound to living cells was quantified using FlowJo software.

**Kinetic analysis of JAM-A-reovirus interactions using BLI**. Virus binding to JAM-A was measured on a BLItz® (Pall ForteBio) biolayer interferometer equipped

with a $Ni^{2+}$-NTA biosensor (Pall ForteBio). After loading the chip in a 10 mM $NiCl_2$ solution for 2 min and running an initial baseline step in milliQ water (1 min), JAM-A (0.2 mg mL$^{-1}$) was immobilized on the exposed $Ni^{2+}$ ions via its C-terminal His$_6$ tag for 5 min until the binding signal reached a plateau (complete saturation of the biosensor). Binding of viral particles (T3SA+ , T3SA− or ISVP; at 16 nM) in the absence or presence of 1 mM Neu5Ac was measured during a 10 min association step after another baseline step (virus buffer for 1 min). Dissociation was monitored directly after the association step for 10 min during which the virus solution was exchanged with virus buffer. Chip can be regenerated several times by exposing the biosensor to 10 mM Glycine pH 1.7 followed by a neutralization buffer (Kinetics Buffer). The resulting sensorgram (binding over time) was processed and fitted with a nonlinear regression approach using an association and then dissociation fit provided by GraphPad Prism. Virus concentration and time at which dissociation was initiated were constrained to constant values of 16 nM and 17 min, respectively. From that fit, $k_{off}$ and $k_{on}$ were extracted and $K_D$ was calculated.

**SPT using dynamic confocal microscopy imaging.** Lec2-JAM-A cells expressing mCherry and CHO-JAM-A cells were co-cultured on a 47-mm glass-bottomed petri dish (WillCo Wells) for 1 or 2 days before the experiment to ensure formation of a confluent monolayer on the day of the experiment. Cells were imaged by laser-scanning confocal microscopy using a Zeiss LSM 880 microscope with a 561 nm laser for mCherry and a 488 nm laser for Alexa Fluor 488 and a ×40 oil objective (NA = 0.95). All experiments were conducted at room temperature with cells maintained in Ham's F12 culture medium and a gas mixture of synthetic air with 5% $CO_2$ at 95% relative humidity that was infused at 0.1 L min$^{-1}$ into the microscopy chamber using a gas humidifier membrane (PermSelect silicone). The humidity was controlled using a humidity sensor (Sensirion). To ensure virus binding (and not internalization), the cells were placed on ice for 30 min before starting the experiment. After finding an area in which both cell types were adjacent (guided by fluorescence), Alexa Fluor 488-labeled T3SA+ or T3SA− viruses (10$^{12}$ particles/mL) diluted in either F12 Ham's culture medium or 1 mM Neu5Ac solution (on ice) were added to the living cells. Fluorescent signals from both dyes (mCherry and Alexa Fluor 488) as well as the signal from the PMT channel were recorded for a ~30 min interval immediately after virus injection at a frame-rate of one image every 13.32 s. During recording, the focus was kept constant on the upper surface of cells. Fluorescence images were exported as 12-bit TIFF files, merged into a movie, and further processed using ImageJ (National Institutes of Health, Bethesda). Trajectories were harvested and analyzed using MTrackJ, an ImageJ plugin to track moving viral particles in the movie and obtain track statistics. The latter were further processed using Origin.

**Antibody staining of T3 reovirus particles on AFM tips.** Individual AFM cantilevers functionalized with virus were placed into wells of a 24-well plate (Corning) and incubated at room temperature for 1 h in 500 µL blocking buffer (PBS with 3% BSA). An antibody against serotype 3 reovirus σ1 protein (9BG5, 0.15 mg mL$^{-1}$)[51] was diluted 1:200 in blocking buffer. Reovirus antibody was prepared by mixing equal volumes of sera from rabbits immunized and boosted with T3D reovirus[52]. The mixed serum was pre-adsorbed on a monolayer of CHO cells to deplete non-specific antibodies. Each cantilever was incubated in 500 µL of the primary antibody solution at room temperature for 1 h. Cantilevers were washed three times with blocking buffer. A secondary antibody solution was prepared by adding a rat anti-mouse IgG2a antibody conjugated to allophycocyanin (APC) fluorophore (Thermo Fisher, catalog # 17-4210-82) at 1:400 dilution in blocking buffer. Cantilevers were incubated in 500 µl of the secondary antibody solution at room temperature for 1 h. Finally, cantilevers were washed three times in PBS and stored at 4 °C in the dark until further use. The cantilevers were imaged using the 488 nm laser line of an inverted confocal microscope (Zeiss LSM 880).

**AFM imaging of reovirus virions adsorbed on HOPG substrate.** An 80 µl droplet of virus solution ( ~10$^9$ particles mL$^{-1}$) was deposited on a freshly cleaved HOPG (highly oriented pyrolytic graphite, NT-MDT instruments) substrate and incubated at room temperature for 15 min. AFM imaging was conducted in the PeakForce Tapping mode using AC40 Biolever mini AFM tips (nominal spring constant 0.1 N m$^{-1}$, Bruker) in PBS buffer. Depending on the desired resolution and scan size, different imaging parameters were used: tip oscillation frequency ranged between 1 and 2 kHz, maximum peak force was 100 pN, scan rates ranged from 0.5 to 2 kHz, peak force amplitudes were between 50 and 100 nm, and resolution was 256 or 512 pixels per line (256 or 512 lines, respectively).

**Reporting summary.** Further information on research design is available in the Nature Research Reporting Summary linked to this article.

## Data availability

The Source data underlying Fig. 2b, c; 3e, f; 4b, c; 5e, f; 6b-e; 7p-s; 8a, f and Supplementary Figs. 5j; 6a-e; 7; 8 are provided as a Source Data file. All other relevant data are available from the corresponding authors upon reasonable request.

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

## Acknowledgements

This work was supported by the Université catholique de Louvain and the Fonds National de la Recherche Scientifique (FRS-FNRS), the Erwin-Schroedinger Fellowship Abroad (FWF Austria, M.K.), and the 'MOVE-IN Louvain' Incoming postdoctoral fellowship programme (A.C.D.). D.A. is a Research Associate at the FNRS. This project received funding from the European Research Council under the European Union's Horizon 2020 research and innovation program (grant agreement no. 758224). Additional support was provided by U.S. Public Health Service awards R01 AI038296 and R01 AI118887 (P.A., C.G., and T.S.D.), UPMC Children's Hospital of Pittsburgh (P.A.), and the Heinz Endowments (T.S.D.). The funders had no role in study design, data collection and analysis, decision to publish, or preparation of the manuscript.

## Author contributions

M.K., P.A., T.S.D., and D.A. conceived the project, planned the experiments, and analyzed the data. M.K. conducted the AFM experiments. P.A. and C.G. engineered the viruses and cell lines and conducted flow cytometry binding experiments. M.K., J.Y., and A.C.D. conducted confocal microscopy experiments. M.K., P.S., and S.G. conducted and analyzed the BLItz experiments. All authors wrote the manuscript.

## Competing interests

M.K. and D.A. have applied for a patent for the use of sialylated glycans in combination with reoviruses (EP19152640.9). The remaining authors declare no competing interests.
