## [Peer Review File · Nature Communications]

Reviewers' comments:

Reviewer #1 (Remarks to the Author):

This study shows the interplay between reovirus binding to cell surface receptors α -SA and JAM-S, highlighting the importance of multivalent interactions and of glycans in strengthening the bond between the reovirus and its target cell. It is largely based on force spectroscopy of the bonds between a reovirus and model and cell surfaces. The study contains a most extensive data set, but in my understanding leaves important questions open, which I believe should be addressed or - in the certainly not impossible case that I have misunderstood/misread things - clarified.

1) A possibly naive question: Why is there no sign in these AFM experiments (on cells) of the reovirus being internalized? What factors are missing that prevent endocytosis?

2) What is the functional relevance of the here presented findings on different forces between reovirus and its receptors on the cell surface? Is the reovirus binding a bottleneck in the process of infection? What evidence do the authors have that the forces and differences in forces are of functional relevance in affecting viral infection?

3) Figs 2h and 3h: Why are there no model surface data in the same loading rate regime ($>10^6$ pN/s) as in the experiments on cells? And since the forces on cells appear so much larger, what is the evidence to support the authors' assumption that they are looking at the same interactions and that the difference is just due to a difference in the number of bonds?

4) Fig. 5p-t show force distributions under different conditions and the authors argue (p. 11) that these differences are significant. Looking at the distributions, I can agree that there are some differences, but these differences appear rather minor. What is the evidence that these differences are sufficiently large to be of functional importance?

5) The adhesion maps are somewhat hard to read. E.g. in Fig. S4, c and d are supposed to demonstrate reproducibility (i.e., similarity) and in g,h and k,l differences. It may be a problem with the graphical quality, but in the manuscript as I have it, these adhesion maps are insufficiently clear to draw such conclusions.

Reviewer #2 (Remarks to the Author):

In this work, Koehler and co-workers exploit atomic force microscopy to study the interactions of single virus particles with both model membranes and living cells. They use mammalian reovirus as a model virus and study its interaction with its attachment factor alpha-SA and with the main reovirus receptor JAM-A. They can show that initial binding of the fiber protein sigma1 to SA promotes the subsequent binding of sigma1 to JAM-A. This stepwise binding model has long been discussed and proposed in the field but was never directly proven. This work directly addresses this question and provides multiple lines of evidences demonstrating this model in a convincing manner. This is a great piece of work with great perspectives not only for reovirus but for virology in general as such method could be exploited for many viruses. Overall, the work is well performed with appropriate statistics and adapted controls. There are a few places where additional controls might be performed to fully support the author statements (see comments below). My main problem with the paper is that it is difficult to read. I understand that there is a limit in space, but the authors should try to make this paper more accessible for non-specialist and non-AFM people. It will be great if they could revise the text and take the readers "by the hand" to drive them smoothly between the different sections (see comments below).

My specific comments are:

Major comments

Do the authors have an explanation why the viruses display a greater number of bindings with the cells compared to the model surface which happens both for SA binding and JAM-A binding? Is it due to the difference in a density of SA and JAM-A on model surfaces compared to cell surfaces? Maybe the author could measure this and correlate this density to their results obtained from WE prediction. This is particularly important for the binding of JAM-A since Lec2 cells express SA (Fig S2). Binding frequencies are much lower when using T3SA- virus but if the author plot the DFS plot using the T3SA- virus (Fig 3h) do they obtain the same number of binding events?

The authors try to make a direct comparison between the structure of sigma1 following binding to SA and the extended structure of sigma1 in an ISVP. I agree that in both configurations the sigma1 protein better binds JAM-A but with the current experiments, it is unclear whether SA binding has the same effect in terms of structural reagent on sigma1 than conversion to ISVP has. The author should test this by testing whether or not binding of ISVP to JAM-A can be increased by treatment with SA.

There is a strong discrepancy between the increased frequency of bindings and increased number of established bounds upon SA treatment between model membranes (Fig 4) and living cells (Fig 5). This might be due to the fact that Lec2 cells are not deficient in SA (as claimed by authors) but have a strong reduction (around 70%) of SA containing glycoproteins (They have un-perturbed levels of CMP-SA). As such, the increased affinity of sigma1 for JAM-A upon SA treatment in living cells should be controlled using the T3SA- strain. This is particularly important since the increased in binding affinity is not as clear than in model membranes. Similarly, what is the increased of frequency of binding upon SA treatment on model membrane. This increased value should be directly compared to the one obtained in living cells. I am wondering whether the limited increased of affinity upon SA treatment in living cells is due to the fact that Lec2 cells have low levels of SA containing glycoproteins explaining the minimum improvement upon SA treatment compared to model membrane. Although, I am convinced that SA induce a structure rearrangement of sigma1 improving its binding affinity to JAM-A (from the model membrane data), this conclusion may have to be tuned down using the Lec2 cells. To better probe the effect of SA on sigma1 binding to Jam-A in cell really lacking SA, the authors could performed the same experiments but on cells which have been pre-treated with neuraminidase to remove all (most) SA for cell surface. This could also be performed in Lec2 cells to further decrease SA levels. I would expect a better effect similar to the one seen in model membranes.

When using single particle tracking (Figure 6) the authors are comparing the diffusion parameters of the viral particles on the cell surface of CHO cells vs. Lec2 cells.

I understand that Lec2 cells are a mutant derived from CHO cells but I am not sure the only difference between these two cell types reside on having less SA at their surfaces and I am wondering whether the authors are now comparing diffusion rate on two different cell types which is hard to interpret. Unless the author could justify that SA is the only difference in Lec2 cells compared to CHO cells, I don't think they can conclude this experiment so sharply. On the other hand, when the authors use Neu5Ac there is a decrease in diffusion rate and increase of number of particle binding, strongly supporting their model. However, full specificity should be controlled by using Neu5Ac on T3SA-, and here again neuraminidase treatment of cells could also be a nice complementary approach.

Minor comments

Line 82-90: Figure S1 and S2 and S3 are mentioned but with very little to no explanations. It is very difficult to know what the authors want us to look at since they only refer the figure as (Sup Fig X). Which panel, which experiemnts? It makes it difficult to understand and follow the very beginning of the MS. This is critical to clarify this beginning and help the reader particularly for people not familiar with the approach. This comment applies for many sections of the text where some re-writing will help the reader follow better the paper

Line 122: the author could add a small conclusion sentence to summarize the biological take home message

Line 184: a small conclusion sentence will help the reader

Figure 4f is not cleared, what are the grey and pink colors?

Reviewer #3 (Remarks to the Author):

The current manuscript deals with force spectroscopy experiments on virus cell binding using atomic force microscopy. Mammalian reoviruses are used and their binding to host cells. This is not only of interest from a fundamental science point of view, but also because reoviruses are studied for possible applications in oncolytic therapies. The cellular surface receptors for this virus are known: α -linked sialic acid and JAM-A. These receptors are bound by the sigma-1 attachment protein of the reovirus. The authors find that initial sialic acid binding leads to enhancement of subsequent JAM-A binding. It is speculated that a conformational change in sigma-1 is responsible for this enhancement.

Below find some specific comments.

The sentence on lines 32-35 in the abstract is not directly clear upon first time reading and this might be reformulated

It is unclear what the bivalent and trivalent interactions are, mentioned on page 6. Are these representing several viral particles? That should be discussed there, not later in the paper. The sentence on lines 121-122 also needs explanation.

line 147: Alexa Flour should be Alexa Fluor

It is not directly clear from the flow of the text how for the cell work the interactions with α -linked sialic acid and JAM-A are separated. How can only the interaction with one receptor be probed and not with the other one? What does this mean for the cells? Can we really compare these experiments if the cells exhibit different surface characteristics?

line 283 characterizing is incorrectly written

line 301, there should be no degree sign if K is used as temperature unit

Retraction velocities are mentioned in the methods for the model surfaces, but loading rates in the figures. The conversion from one into the other should be mentioned for these experiments.

An important control is the check whether the structure of the particles is as expected. Suppl. fig 3a,b shows images of the reoviruses, but these images are worrisome as it seems that the particles have quite aberrant morphology. This indicates something is wrong.

Point-by-Point Response to the Reviewers Comments

Reviewer #1 (Remarks to the Author):

This study shows the interplay between reovirus binding to cell surface receptors α -SA and JAM-A, highlighting the importance of multivalent interactions and of glycans in strengthening the bond between the reovirus and its target cell. It is largely based on force spectroscopy of the bonds between a reovirus and model and cell surfaces. The study contains a most extensive data set, but in my understanding leaves important questions open, which I believe should be addressed or - in the certainly not impossible case that I have misunderstood/misread things - clarified.

Authors: Thank you for your encouraging and constructive comments. Below we have explained point-by-point how we have addressed your questions and concerns to strengthen our manuscript.

1) A possibly naive question: Why is there no sign in these AFM experiments (on cells) of the reovirus being internalized? What factors are missing that prevent endocytosis?

Authors: The referee questions whether there is evidence of reovirus internalization in our experiments. In a previous study, Ehrlich *et al.* showed that during the early stages of reovirus binding to the cell surface (between 280-1500 s after the first contact of the virus with the cell), reovirus particles are essentially stationary with an average lateral displacement of 10-30 nm.s⁻¹.¹ In AFM studies, the contact time between the virus and cell surface is in the ~ millisecond range. Therefore, the probability of observing internalization events is extremely low. In addition, the virus is stably linked to an AFM tip *via* a covalent bond. Together with the control experiments shown in Supplementary Figs. 4 and 5, we are certain that the virus is stably attached to the AFM tip and not being internalized during the interval of measurement.

2) What is the functional relevance of the here presented findings on different forces between reovirus and its receptors on the cell surface? Is the reovirus binding a bottleneck in the process of infection? What evidence do the authors have that the forces and differences in forces are of functional relevance in affecting viral infection?

Authors: We quantified the forces established between individual reovirus virions and receptors. Extraction of the binding free-energy landscape for each receptor (sialic acid [SA] and

JAM-A) enabled us to determine the number of bonds established between the virus and the cell surface. The number of bonds and diffusion are intimately linked.^{2,3} In the case of reovirus, after landing on the cell surface and binding to receptors, the virus adheres to a confined location from which it will be endocytosed in a signal-induced manner.⁴ Therefore, in the case of reovirus, we think virions maximize the number of bonds to reduce their lateral diffusion. In this context, the study and extraction of the number of virus-receptor bonds is relevant to understand initiation of the infection process.

3) Figs 2h and 3h: Why are there no model surface data in the same loading rate regime ($>10^6$ pN/s) as in the experiments on cells? And since the forces on cells appear so much larger, what is the evidence to support the authors' assumption that they are looking at the same interactions and that the difference is just due to a difference in the number of bonds?

Authors: In the case of specific interactions, the Bell-Evans model^{5,6} predicts that far-from-equilibrium, the rupture force (*e.g.*, strength) of the ligand-receptor bond is proportional to the logarithm of the loading rate (LR), which is the rate at which the force is applied to the system ($LR=\Delta F/\Delta t$). However, the extraction of kinetic parameters requires the fit to be extrapolated to $LR=0$. Therefore, rupture forces are analyzed over a wide range of LRs. As the rupture force increases linearly with the logarithm of the loading rate, there is no need to overlap both measurements (although data overlap in our case). In our experiments using living cells, the data were acquired with the FD-based AFM mode that operates at higher frequency (hence higher LR), enabling us to record more pixels within the map (256 by 256 pixels in this mode compared with 32 by 32 pixels for the mode used for the model surfaces). This approach results in an increase in the lateral resolution of the map and facilitates localization of receptors on the cell. The number of bonds is predicted by the William-Evans calculation⁷, which also shows a linear dependency on the LR. In addition, the various controls performed confirm that we are looking at the same interactions on both model surfaces and cells.

4) Fig. 5p-t show force distributions under different conditions and the authors argue (p. 11) that these differences are significant. Looking at the distributions, I can agree that there are some differences, but these differences appear rather minor. What is the evidence that these differences are sufficiently large to be of functional importance?

Authors: The reviewer questions whether differences in the number of bonds observed between the virus and the cells are significant. First, we emphasize that we obtained at least 700 data points per condition to construct the force histograms and the subsequent multi-peak Gaussian fitting. These data were obtained using at least 15 different cells from 5 independent experiments. In addition, we conducted a statistical test on the analysis of the established number of bonds (shown in Supplementary Table 1) indicating that the differences are significant under the tested conditions.

To determine whether the number of bonds formed is functionally important, we also conducted single virus particle tracking experiments (Fig. 6b-f, Supplementary Fig. 9). These experiments clearly show a significant effect of free SA addition on the binding and diffusion potential of reovirus particles on CHO-JAM-A and Lec2-JAM-A cells. Together, these data demonstrate that differences in the number of bonds established is sufficiently large to be functionally important and strengthen our conclusion that glycans mediate an enhancement of reovirus receptor binding.

For clarification, the number of data points to generate the force histograms is now included in the Fig. 5 legend (see lines 831-832).

5) The adhesion maps are somewhat hard to read. E.g. in Fig. S4, c and d are supposed to demonstrate reproducibility (i.e., similarity) and in g,h and k,l differences. It may a problem with the graphical quality, but in the manuscript as I have it, these adhesion maps are insufficiently clear to draw such conclusions.

Authors: To alleviate the reviewer's concern, we have revised Figs. 2 and 3 as well as Supplementary Figs. 4 and 5 and enlarged the pixel size by two-fold. With this improvement in the visibility of the maps, the reproducibility of the experiments highlighted in Supplementary Fig. 4c,d and the differences in g, h and k, l are now more clear. In addition, we will make sure to provide the figures at the highest possible resolution.

Reviewer #2 (Remarks to the Author):

In this work, Koehler and co-workers exploit atomic force microscopy to study the interactions of single virus particles with both model membranes and living cells. They use mammalian reovirus as a model virus and study its interaction with its attachment factor alpha-SA and with the main reovirus receptor JAM-A. They can show that initial binding of the fiber protein sigma1 to SA promotes the subsequent binding of sigma1 to JAM-A. This stepwise binding model has long been discussed and proposed in the field but was never directly proven. This work directly addresses this question and provides multiple lines of evidences demonstrating this model in a convincing manner. This is a great piece of work with great perspectives not only for reovirus but for virology in general as such method could exploited for many viruses. Overall, the work is well performed with appropriate statistics and adapted controls. There are a few places where additional controls might be performed to fully support the author statements (see comments below). My main problem with the paper is that it is difficult to read. I understand that there is limit in space, but the authors should try to make this paper more accessible for non-specialist and non-AFM people. It will be great if they could revise the text and take the readers “by the hand” to drive them smoothly between the different sections (see comments below).

Authors: Thank you for your encouraging and constructive comments. Below we have explained point-by-point how we have addressed these comments in our revision. In particular, we have conducted all of the suggested control experiments to more fully support our conclusions and endeavored to make the text more accessible to a wider audience.

Major comments

1) Do the authors have an explanation why the viruses display a greater number of bindings with the cells compared to the model surface which happens both for SA binding and JAM-A binding? Is it due to the difference in a density of SA and JAM-A on model surfaces compared to cell surfaces? Maybe the author could measure this and correlate this density to their results obtained from WE prediction. This is particularly important for the binding of JAM-A since Lec2 cells express SA (Fig S2). Binding frequencies are much lower when using T3SA- virus but if the author plot the DFS plot using the T3SA- virus (Fig 3h) do they obtain the same number of binding events?

Authors: In the first part of this comment, the referee asked why the viruses display a greater number of established bonds on cell-surface receptors compared with results from experiments using model surfaces. A direct comparison of binding frequencies in these experiments is not straightforward. First, the experiments were conducted using different AFM modes (FV vs. PF-QNM) that differ in their contact times (~ 25 ms in FV vs. ~ 1 ms in PF-QNM) and therefore directly influence the binding probability. Second, the stiffness of the model surfaces (stiff substrate covered with purified proteins vs. living cells) influences the contact area and consequently the binding probability. Finally, the density of SA and JAM-A most probably differs

on the model substrates compared with living cells, which in turn affects the binding probability. Also, on living cells, we expect the receptors not to be homogeneously distributed.

Regarding the interaction of virus with cells, we do not observe significant differences between T3SA+ and T3SA- binding to Lec2-JAM-A cells (Fig. 3i) with binding frequencies of $3.5 \pm 1.1\%$ and $4.0 \pm 1.2\%$, respectively. It is also important to mention that the differences in binding frequencies are not important. On model surfaces, the key point is to be able to quantify the binding force at the level of single-molecule rupture events, enabling us to extract the kinetic parameters using the Bell-Evans model and to “model” multiple rupture events using the Williams-Evans prediction. This calibrated system is in turn used to evaluate the multivalence of the bond established on cells. Nevertheless, we extracted the data recorded with T3SA- virus and Lec2-JAM-A cells and overlaid it with the data obtained using the model surface (Supplementary Fig. 5j). We observed binding events with a similar number of uncorrelated bonds established in parallel on cell surface (between I and IV bonds with a maximum at II bonds).

Figure S5i,j Control experiments for studying the contribution of JAM-A in reovirus binding to living cells. (i-j) DFS analysis of T3SA- interactions with JAM-A extracted from adhesion areas on Lec2-JAM-A cells. (i) Cartoon of the experiment. (j) DFS plot of T3SA- interactions with JAM-A on model surfaces (grey circles, taken from Fig. 3b – lower panel) and living cells (red dots). Histogram of the force distribution observed on cells fitted with a multi-peak Gaussian distribution (n = 620) is shown on the side.

These new data further support our conclusion that we indeed probe specific interactions between the virus and JAM-A rather than between the virus and remaining SA glycans on Lec2-JAM-A cells.

To provide further support for this contention, we conducted an additional control experiment in which neuraminidase was used to remove residual SA on Lec2-JAM-A cells prior to binding studies.

Please see changes in the manuscript on lines 208-214 and Supplementary Fig. 5.

2) The authors try to make a direct comparison between the structure of sigma1 following binding to SA and the extended structure of sigma1 in an ISVP. I agree that in both configurations the sigma1 protein better binds JAM-A but with the current experiments, It is unclear whether SA binding has the same effect in term of structural reagent on sigma1 than conversion to ISVP has. The author should test this by testing whether or not binding of ISVP to JAM-A can be increased by treatment with SA.

Authors: We thank the reviewer for suggesting this additional control experiment to strengthen our hypothesis that SA binding appears to mediate structural rearrangement of $\sigma 1$ at the viral particle level analogous to the $\sigma 1$ conformer present in ISVPs. We tested whether binding of ISVPs to JAM-A is increased by incubation with SA. The results are shown in revised Fig. 4g and Supplementary Fig. 6d and summarized in Supplementary Fig. 6e. We did not detect alteration of ISVP-JAM-A binding after incubation with free SA, strengthening our conclusion that free glycans mediate an enhancement in receptor binding by reovirus virions due to a conformational change in $\sigma 1$ similar to that observed during conversion of virions to ISVPs.

Figure S6e | Testing the effect of free SA compounds on T3SA- binding to JAM-A. (e) Box plot of BF observed for JAM-A-T3SA+ (left panel), JAM-A-T3SA- (middle panel) and JAM-A-T3SA+ ISVP (right panel) interactions, without adding SA compounds (grey for T3SA+, white for T3SA-, blue for T3SA+ ISVP) and after adding Neu5Ac (red), LSTa (yellow), or LNnT (green), as well as after injection of 10 $\mu\text{g/ml}$ JAM-A Ab as a receptor-blocking reagent (dashed lines in the respective boxes). The observed reduction in binding frequency in the presence of JAM-A Ab verifies the specificity of observed interactions.

Please see changes in the manuscript on lines 244-251 as well as Fig. 4 and Supplementary Fig. 6.

3) There is a strong discrepancy between the increased frequency of bindings and increased number of established bounds upon SA treatment between model membranes (Fig 4) and living cells (Fig 5). This might be due to the fact that Lec2 cells are not deficient in SA (as claimed by authors) but have a strong reduction (around 70%) of SA containing glycoproteins (They have un-perturbed levels of CMP-SA). As such, the increased affinity of sigma1 for JAM-A upon SA treatment in living cells should be controlled using the T3SA- strain. This is particularly important since the increased in binding affinity is not as clear than in model membranes. Similarly, what is the increased of frequency of binding upon SA treatment on model membrane. This increased value should be directly compared to the one obtained in living cells. I am wondering whether the limited increased of affinity upon SA treatment in living cells is due to the fact that Lec2 cells have low levels of SA containing glycoproteins explaining the minimum improvement upon SA treatment compared to model membrane. Although, I am convinced that SA induce a structure rearrangement of sigma1 improving its binding affinity to JAM-A (from the model membrane data), this conclusion may have to be tuned down using the Lec2 cells. To better probe the effect of SA on sigma1 binding to Jam-A in cell really lacking SA, the authors could perform the same experiments but on cells which have been pre-treated with neuraminidase to remove all (most) SA for cell surface. This could also be performed in Lec2 cells to further decrease SA levels. I would expect a better effect similar to the one seen in model membranes.

Authors: For the discrepancy between model membranes (Fig. 4) and living cells (Fig. 5) in the increased frequency of binding and increased number of established bonds following SA treatment, we refer to our response to comment #1.

Regarding the suggestion to further control for the possibility of increased affinity of $\sigma 1$ for JAM-A on living cells following SA treatment of the T3SA- strain, we think that these experiments will not yield useful information. Indeed, we extensively studied the binding behavior of T3SA- to JAM-A following SA treatment on model surfaces (Fig. 4 and Supplementary Fig. 6). We show that the incubation of T3SA- with sialylated glycans does not influence its binding to JAM-A or the establishment of multivalent interactions, demonstrating conclusively that the SA binding site in T3SA+ is responsible for our observations.

Regarding the fact that Lec2 cells retain low levels of SA expression, we thank the reviewer for this suggestion, and we conducted the suggested additional control experiments. We incubated Lec2 cells with neuraminidase to remove residual SA. As described in the Materials and Methods section, the medium was removed, and cells were washed twice with 2 mL of PBS, treated with *Arthrobacter ureafaciens* neuraminidase (Sigma-Aldrich) at a final concentration of 40 mUnit/mL in PBS for 1 h, and washed with 2 mL PBS. AFM experiments were conducted using cells maintained in Mem α , nucleosides medium without FBS to limit SA recovery.

Using our AFM strategy, we monitored the influence of basal levels of SA on Lec2 cells by probing T3SA+ binding before and after neuraminidase treatment. We also tested the effect of

Neu5Ac incubation after neuraminidase treatment and blocked JAM-A binding using JAM-A Ab (see Supplementary Figure S8).

Figure S8 | Monitoring the effect of SA addition on reovirus binding to living cells after neuraminidase treatment.

(a) Cartoon of the injection experiment highlighting that Lec2 cells are fluorescently labelled and the running order of the injections. (b) FD-based AFM height image (25 μm x 25 μm fluorescent image of the cells is shown in inset) and corresponding adhesion channels, acquired first in growth medium (c) followed by scanning the same area after neuraminidase treatment (e) to remove remaining SAs on the cell surface. A slight decrease ($P < 0.01$) of adhesion events is observed, indicating that the NA treatment removed potential remaining SAs on the cell surface. (d,f) Enlarged images of adhesion maps recorded on Lec2-JAM-A cells (dashed square in adhesion map). The upper images display the lower force range (300 to 400 pN), whereas the lower images display the higher force range (400 to 500 pN), with significantly fewer adhesion events before and after NA treatment. The frequency of adhesion events is indicated. After NA treatment, free Neu5Ac (1 mM) was added, and the same area scanned again (g). Enlarged images of adhesion maps recorded on Lec2-JAM-A cells (dashed square in adhesion map and similar areas as in c,e show more adhesion events in the high force range upon injection of sialylated glycan. This

result is in line with the experiment carried out on cells without NA treatment (Fig. 5a-e). In a fourth consecutive step, the same area was scanned after injection of 10 µg/ml JAM-A Ab (I, j) to block cell-surface JAM-A molecules. A significant reduction of adhesion events is observed. All AFM images were acquired using an oscillation frequency of 0.25 kHz and amplitude of 750 nm under cell culture conditions. Experiments were repeated 3-5 times. For clarity and better visibility, the pixel size in the adhesion images were enlarged by a factor 2. (k) Box plot of the BF observed for T3SA+ virions first without treatment (grey), followed by NA treatment (blue), addition of free Neu5Ac (red) and finally injection of JAM-A Ab (brown). Data are representative of at four independent experiments. **, $P < 0.01$; ***, $P < 0.0001$; determined by two-way ANOVA corrected for multiple comparisons using Tukey's test in GraphPad Prism or Origin.

As shown in Supplementary Fig. 8, we observed a slight decrease (from $4.1 \pm 0.3\%$ to $3.2 \pm 0.2\%$; $P < 0.01$) in the adhesion events following neuraminidase treatment, indicating that although neuraminidase removed residual SA on the Lec2 cell surface, the SA contribution to Lec2 binding is modest. The final outcome of this experiment is similar to the experiments conducted without neuraminidase treatment (Fig. 5 and Supplementary Fig. 7).

Please see changes in the manuscript on lines 259-269, the new section in the Materials and Methods on lines 507-517, and the new Supplementary Fig. 8.

4) When using single particle tracking (Figure 6) the authors are comparing the diffusion parameters of the viral particles on the cell surface of CHO cells vs. Lec2 cells. I understand that Lec2 cells are a mutant derived from CHO cells but I am not sure the only difference between these two cell types reside on having less SA at their surfaces and I am wondering whether the authors are now comparing diffusion rate on two different cell types which is hard to interpret. Unless the author could justify that SA is the only difference in Lec2 cells compared to CHO cells, I don't think they can conclude this experiment so sharply. On the other hand, when the authors use Neu5Ac there is a decrease in diffusion rate and increase of number of particle binding, strongly supporting their model. However, full specificity should be controlled by using Neu5Ac on T3SA-, and here again neuraminidase treatment of cells could also be a nice complementary approach.

Authors: We agree with the reviewer that we cannot claim that SA expression is the only difference in Lec2 cells relative to CHO cells. Lec2 cells have a deletion in the coding region of the CMP-SA transporter gene, *slc35a1*, and are therefore deficient in transporting SA to the Golgi for conjugation onto proteins. Since Lec2 cells were engineered by chemical mutagenesis of CHO cells, it is possible that other mutations are present. However, since we study interactions between reovirus and cell-surface SA and JAM-A as well as glycan-mediated enhancement of reovirus receptor binding with direct internal controls (co-culture of Lec2-JAM-A mCherry and CHO-JAM-A cells), we are not able to use neuraminidase treatment to remove potential remaining SA on Lec2-JAM-A cells. This treatment also would remove SA on CHO-JAM-A cells, which would preclude assessment of the specificity and interplay of the two binding

partners on reovirus interaction with cells. However, we followed the reviewer's suggestion and conducted an additional experiment to fully control for SA specificity by testing the effect of Neu5Ac incubation on T3SA- binding. The new results were added in the revised Fig. 6 as well as Supplementary Fig. 9 (previously Supplementary Fig. 8). We found that the mean travelled distance and mean velocity of T3SA- virions were comparable in the presence (light red bars) and absence (white bars) of Neu5Ac. In addition, the amount of bound T3SA- is almost identical under both conditions and significantly differs from the effect of incubation of T3SA+ in the presence and absence of Neu5Ac. These new results further support our conclusion that glycan engagement enhances reovirus receptor binding.

Please see changes in the manuscript on lines 299-300.

Minor comments

5) Line 82-90: Figure S1 and S2 and S3 are mentioned but with very little to no explanations. It is very difficult to know what the authors want us to look at since they only refer the figure as (Sup Fig X). Which panel, which experiments? It makes it difficult to understand and follow the very beginning of the MS. This is critical to clarify this beginning and help the reader particularly for people not familiar with the approach. This comment applies for many sections of the text where some re-writing will help the reader follow better the paper

Authors: Due to the extensive amount of data required to support our conclusions and the limited space for text and figures in the manuscript, we had to shift many control experiments to the supplement. We have revised the manuscript to make it easier to follow specific panels.

Please see changes in the manuscript on lines 86-98.

6) Line 122: the author could add a small conclusion sentence to summarize the biological take home message

Authors: We now added a conclusion sentence for the biological take-home message: "Thus, our *in vitro* experiments confirm that T3SA+ virions specifically interact with α -SA glycans and that virions rapidly (in the ms range) establish multivalent bonds with α -SA glycans, presumably providing the virion with its first stable anchorage to the cell surface."

Please see changes in the manuscript on lines 133-136.

7) Line 184: a small conclusion sentence will help the reader

Authors: We added a conclusion sentence: "Together, these results reveal that T3SA+ establishes stable interactions with JAM-A independent of SA engagement".

Please see changes in the manuscript on lines 201-203.

8) Figure 4f is not cleared, what are the grey and pink colors?

Authors: We have added an additional explanation for the colors in the Figure 4 legend.

Reviewer #3 (Remarks to the Author):

The current manuscript deals with force spectroscopy experiments on virus cell binding using atomic force microscopy. Mammalian reoviruses are used and their binding to host cells. This is not only of interest from a fundamental science point of view, but also because reoviruses are studied for possible applications in oncolytic therapies. The cellular surface receptors for this virus are known: α -linked sialic acid and JAM-A. These receptors are bound by the sigma-1 attachment protein of the reovirus. The authors find that initial sialic acid binding leads to enhancement of subsequent JAM-A binding. It is speculated that a conformational change in sigma-1 is responsible for this enhancement. Below find some specific comments.

Authors: Thank you for your encouraging comments. Below we explain point-by-point how we have addressed the concerns in our revised manuscript.

1) The sentence on lines 32-35 in the abstract is not directly clear upon first time reading and this might be reformulated

Authors: We have reformulated the text on lines 32-36 in the abstract, which now reads: “The enhanced JAM-A binding by virions following α -SA engagement is comparable to JAM-A binding by infectious subviral particles (ISVPs) in the absence of α -SA. Since ISVPs are reovirus disassembly intermediates that have an extended σ 1 conformer, this finding suggests that α -SA binding triggers a conformational change in σ 1.”

2) It is unclear what the bivalent and trivalent interactions are, mentioned on page 6. Are these representing several viral particles? That should be discussed there, not later in the paper. The sentence on lines 121-122 also needs explanation.

Authors: We agree that the chronological order in which we discussed the nature of multivalent interactions might lead to confusion. We have revised it accordingly in the manuscript.

Please see changes in the manuscript on lines 150-157.

3) line 147: Alexa Flour should be Alexa Fluor

Authors: We have now corrected this typo.

Please see change in the manuscript on line 163.

4) It is not directly clear from the flow of the text how for the cell work the interactions with α -linked sialic acid and JAM-A are separated. How can only the interaction with one receptor be probed and not with the other one? What does this mean for the cells? Can we really compare these experiments if the cells exhibit different surface characteristics?

Authors: Thank you for this relevant question. Here we need to look at three different situations.

First, we checked interactions of reovirus on independently prepared SA (Fig. 2a-c) and JAM-A (Fig. 3a-c) model surfaces to develop some initial conclusions about the binding behavior, kinetic parameters, affinity and avidity, and number of established bonds.

Following these experiments, we moved to studies using living cells in which we defined the contribution of JAM-A and SA in reovirus binding to different mixtures of cells in a physiologically relevant context. (i) The SA contribution was studied using a co-culture of JAM-A-lacking Lec2-cells (which are deficient in SA) and fluorescently labelled CHO-cells (which express SA) (Fig. 2d-i). The JAM-A contribution was studied using a co-culture of SA-lacking fluorescently labelled Lec2 and Lec2-JAM-A cells (Fig. 3d-i). Using these different cell mixtures, we are certain that we are only investigating interactions with one receptor at a time. In addition, guided by fluorescence, we have a direct internal control for virus binding to the respective cell-surface receptors, as also shown in several control experiments (Supplementary Fig. 4 and 5). Moreover, we can compare findings made in studies of CHO and Lec2 cells, since Lec2 cells were recovered following chemical mutagenesis of CHO cells. Lec2 cells have a deletion in the coding region of the CMP-SA transporter gene, *slc35a1*, and are therefore deficient in transporting SA to the Golgi for addition to proteins. Consequently, Lec2 cells display cell-surface characteristics similar to CHO cells, except for their deficiency in cell-surface SA. To bridge the gap between experiments using model surfaces and living cells, and to validate that the binding events observed using cells are indeed interactions between the virion and the respective cell-surface receptors, we extensively analyzed their kinetic properties. We extracted the force and loading rate from the adhesive curves recorded between virus and cell. We analyzed the force distribution and overlaid the mean rupture forces (as well as single data points [always shown in red]) along with the loading rates on the DFS plot previously obtained using model surfaces (always shown in grey). As can be observed on the DFS plots presented in Fig. 2 and 3, the data obtained in experiments using cells are well aligned with the Bell-Evans fit / Williams-Evans prediction obtained for the interactions measured between purified receptors (SA or JAM-A) and the virus. The good agreement between the datasets corroborates our conclusion that the interactions measured using SA- or JAM-A-expressing cells (together with all of the controls) represent specific interactions with the virus.

We have clarified the text in the revised manuscript to make these points clear.

5) line 283 characterizing is incorrectly written

Authors: We have now corrected this error.

Please see change in the manuscript on line 321.

6) line 301, there should be no degree sign if K is used as temperature unit

Authors: We have removed the degree sign.

7) Retraction velocities are mentioned in the methods for the model surfaces, but loading rates in the figures. The conversion from one into the other should be mentioned for these experiments.

Authors: In the revised manuscript, we have added the conversion from retraction or pulling velocities to loading rates in the methods section for the model surfaces (line 459-461).

“The pulling velocity (v) and loading rate (LR) can be related as follows: $LR = \Delta F/\Delta t = k_{\text{eff}} \cdot v$, where $\Delta F/\Delta t$ being the applied force over time, and k_{eff} the effective spring constant of the system.”

8) An important control is the check whether the structure of the particles is as expected. Suppl. fig 3a,b shows images of the reoviruses, but these images are worrisome as it seems that the particles have quite aberrant morphology. This indicates something is wrong.

Authors: We apologize for the poor quality of the topography image and the concerns raised about the integrity of the virus. The initial experiment was conducted in air, with the virus adsorbed on a pre-treated Ni^{2+} mica substrate, which might lead to aberrant morphology of the particles. To validate the integrity and structure of the viral particles used in our study, we raster-scanned the virus coupled to HOPG substrates under physiologically relevant conditions (*i.e.*, in buffer). As shown in the revised Supplementary Fig. 3, the reovirus particles are round, approximately 80-100 nm in diameter, and display characteristic surface protrusions of the outer-capsid proteins.⁸ We have revised the text about the AFM imaging of reovirus virions in the Materials and Methods section accordingly.

References

- 1 Ehrlich, M. *et al.* Endocytosis by random initiation and stabilization of clathrin-coated pits. *Cell* **118**, 591-605 (2004).
- 2 Ewers, H. *et al.* Single-particle tracking of murine polyoma virus-like particles on live cells and artificial membranes. *Proceedings of the National Academy of Sciences* **102**, 15110-15115 (2005).
- 3 Ewers, H. *et al.* GM1 structure determines SV40-induced membrane invagination and infection. *Nat. Cell Biol.* **12**, 11 (2010).
- 4 Boulant, S., Stanifer, M. & Lozach, P. Y. Dynamics of virus-receptor interactions in virus binding, signaling, and endocytosis. *Viruses* **7**, 2794-2815 (2015).
- 5 Evans, E. & Ritchie, K. Dynamic strength of molecular adhesion bonds. *Biophys. J.* **72**, 1541-1555 (1997).
- 6 Evans, E. A. & Calderwood, D. A. Forces and bond dynamics in cell adhesion. *Science* **316**, 1148-1153 (2007).
- 7 Evans, E. & Williams, P. *Physics of bio-molecules and cells. Physique des biomolécules et des cellules* Ch. Dynamic force spectroscopy, 145-204 (Springer, 2002).
- 8 Dryden, K. A. *et al.* Early steps in reovirus infection are associated with dramatic changes in supramolecular structure and protein conformation: analysis of virions and subviral particles by cryoelectron microscopy and image reconstruction. *J. Cell Biol.* **122**, 1023-1041 (1993).

Reviewers' comments:

Reviewer #1 (Remarks to the Author):

The authors have addressed several of my comments and improved the manuscript. Yet I would recommend that the authors revise their submission to include the (perfectly fine) responses to my first two questions in the manuscript, instead of only in the rebuttal, as I would guess that other readers might have the same questions. Specifically, this applies to my (previous) comments:

1) A possibly naive question: Why is there no sign in these AFM experiments (on cells) of the reovirus being internalized? What factors are missing that prevent endocytosis?

2) What is the functional relevance of the here presented findings on different forces between reovirus and its receptors on the cell surface? Is the reovirus binding a bottleneck in the process of infection? What evidence do the authors have that the forces and differences in forces are of functional relevance in affecting viral infection?

More problematic, however, remains my third question, which has essentially remained unanswered:

3) Figs 2h and 3h: Why are there no model surface data in the same loading rate regime ($>10^6$ pN/s) as in the experiments on cells? And since the forces on cells appear so much larger, what is the evidence to support the authors' assumption that they are looking at the same interactions and that the difference is just due to a difference in the number of bonds?

I understand the Bell-Evans model and extrapolation to zero loading rate, but the point is that the cell data only cover one order-of-magnitude of loading rates, which makes such an extrapolation questionable on the cell data per se. It would be beneficial if the authors can make explicit what is the evidence they are here looking at the same interactions (as on the model surface) and that the difference is just due to a difference in the number of bonds. The current rebuttal is rather vague and does not provide a response to the question what the evidence is: "The number of bonds is predicted by the William-Evans calculation[7], which also shows a linear dependency on the LR. In addition, the various controls performed confirm that we are looking at the same interactions on both model surfaces and cells." I am sure the authors can (and should) provide a more convincing and more specific answer, and also adjust the manuscript accordingly.

Reviewer #2 (Remarks to the Author):

I apologize to the authors for the delay in evaluating their revised manuscript. I had multiple deadlines on my side and was traveling for the last two weeks.

I carefully read their revised manuscript which i found significantly improved in term of readability. The addition of the requested controls further improves this work and also help to drive the reader smoothly in this Complex study

I have no more comments for the authors and would like to congratulate them for this work

Steeve Boulant

Reviewer #3 (Remarks to the Author):

The authors of the manuscript have successfully addressed the points that I raised. The new data and the adjustment + update of the text have made the manuscript overall stronger. It reads more logical and clearer now.

Point-by-Point Response to the Editors and Reviewer Comment

Reviewer #1 (Remarks to the Author):

The authors have addressed several of my comments and improved the manuscript. Yet I would recommend that the authors revise their submission to include the (perfectly fine) responses to my first two questions in the manuscript, instead of only in the rebuttal, as I would guess that other readers might have the same questions. Specifically, this applies to my (previous) comments.

Authors: Thank you for your encouraging and constructive comments. Below, we have explained point-by-point how we have addressed these comments in our second revision. In particular, we have included all of the suggested changes in the revised manuscript.

1) A possibly naive question: Why is there no sign in these AFM experiments (on cells) of the reovirus being internalized? What factors are missing that prevent endocytosis?

2) What is the functional relevance of the here presented findings on different forces between reovirus and its receptors on the cell surface? Is the reovirus binding a bottleneck in the process of infection? What evidence do the authors have that the forces and differences in forces are of functional relevance in affecting viral infection?

Authors: We agree with the reviewer that these questions might be of interest to other readers. We have now added our responses to question 1 on lines 149-154 and to question 2 on lines 339-343.

3) Figs 2h and 3h: Why are there no model surface data in the same loading rate regime ($>10^6$ pN/s) as in the experiments on cells? And since the forces on cells appear so much larger, what is the evidence to support the authors' assumption that they are looking at the same interactions and that the difference is just due to a difference in the number of bonds?

I understand the Bell-Evans model and extrapolation to zero loading rate, but the point is that the cell data only cover one order-of-magnitude of loading rates, which makes such an extrapolation questionable on the cell data per se. It would be beneficial if the authors can make explicit what is the evidence they are here looking at the same interactions (as on the model surface) and that the difference is just due to a difference in the number of bonds. The current rebuttal is rather vague and does not provide a response to the question what the evidence is: "The number of bonds is predicted by the William-Evans calculation[7], which also shows a linear dependency on the LR. In addition, the various controls performed confirm that

we are looking at the same interactions on both model surfaces and cells." I am sure the authors can (and should) provide a more convincing and more specific answer, and also adjust the manuscript accordingly.

Authors: While the loading rates used in experiments conducted with cells and model surfaces are different, they still overlap (in the range 10^5 - 10^6 pN/s) enabling the direct comparison of forces.

We also conducted several control experiments to provide strong evidence that we are studying the same, specific interactions on both cells and model surfaces. The first control is the use of Lec2 cells, which lack α -SA on the cell-surface. We show a significant drop in the binding frequency of reovirus strain T3SA+ to these cells. Next, we conducted two similar controls using cells and model surfaces: (i) probing the same CHO-Lec2 cell mixture first with T3SA+ on the AFM tip and then with a T3SA- on the tip (**Fig. 2i** and **Supplementary Fig. 4e-h**) and (ii) blocking specific virus-glycan interactions using 1 mM Neu5Ac (**Fig. 2i** and **Supplementary Fig. 4i-l**). The results gathered from these experiments are very similar, as can be seen in Fig. 2b and Fig. 2i. There is a significant decrease in the binding frequency on probing with T3SA- virions, which do not engage α -SA, or when we inject free α -SA in solution, confirming that we are probing the same interaction in both cases. We have revised the text (lines 170 – 174) in the manuscript to explain the observed differences.